# L-Type Ca^2+^ Channel Inhibition Rescues the LPS-Induced Neuroinflammatory Response and Impairments in Spatial Memory and Dendritic Spine Formation

**DOI:** 10.3390/ijms232113606

**Published:** 2022-11-06

**Authors:** Jieun Kim, Seong Gak Jeon, Ha-Ram Jeong, HyunHee Park, Jae-Ick Kim, Hyang-Sook Hoe

**Affiliations:** 1Department of Neural Development and Disease, Korea Brain Research Institute (KBRI), 61, Cheomdan-ro, Dong-gu, Daegu 41062, Korea; 2Department of Biological Sciences, Ulsan National Institute of Science and Technology (UNIST), Ulsan 44919, Korea; 3Department of Brain and Cognitive Science, Daegu Gyeongbuk Institute of Science & Technology (DGIST), 333, Techno Jungang-Daero, Hyeonpung-eup, Dalseong-gun, Daegu 42988, Korea

**Keywords:** felodipine, LPS, gliosis, neuroinflammation, Ca^2+^ channel blocker, spatial memory

## Abstract

Ca^2+^ signaling is implicated in the transition between microglial surveillance and activation. Several L-type Ca^2+^ channel blockers (CCBs) have been shown to ameliorate neuroinflammation by modulating microglial activity. In this study, we examined the effects of the L-type CCB felodipine on LPS-mediated proinflammatory responses. We found that felodipine treatment significantly diminished LPS-evoked proinflammatory cytokine levels in BV2 microglial cells in an L-type Ca^2+^ channel-dependent manner. In addition, felodipine leads to the inhibition of TLR4/AKT/STAT3 signaling in BV2 microglial cells. We further examined the effects of felodipine on LPS-stimulated neuroinflammation in vivo and found that daily administration (3 or 7 days, i.p.) significantly reduced LPS-mediated gliosis and COX-2 and IL-1β levels in C57BL/6 (wild-type) mice. Moreover, felodipine administration significantly reduced chronic neuroinflammation-induced spatial memory impairment, dendritic spine number, and microgliosis in C57BL/6 mice. Taken together, our results suggest that the L-type CCB felodipine could be repurposed for the treatment of neuroinflammation/cognitive function-associated diseases.

## 1. Introduction

Neuroinflammation is the result of an immune response in the central nervous system (CNS), including the brain and spinal cord. Transient and appropriate levels of neuroinflammation provide benefits such as neuroprotection, neuroplasticity, and tissue remodeling [1,2]. However, uncontrolled, excessive, and chronic neuroinflammation not only damages neurons and abolishes plasticity but also accelerates pathological neuroinflammation due to the influx of peripheral immune factors via disruption of the blood–brain barrier (BBB) [1,3]. Chronic neuroinflammation has been implicated in transient CNS damage, chronic autoimmune diseases, and neurodegenerative diseases [4,5,6,7]. Neuroinflammatory responses are regulated by various proinflammatory cytokines and secondary messengers (prostaglandins and nitric oxide), which are mediated by neurons, microglia, and astrocytes residing in the CNS [8,9].

Microglia, which are macrophage-like immune cells residing in the CNS, play a pivotal role in neuroinflammation [10]. Microglia secrete neuroprotective factors, remove pathogens or debris, and are activated by stimuli such as pathogens and ATP leaks induced by infection or lesions [11,12]. Activated microglia mediate neuroinflammation by promoting proinflammatory cytokine secretion, chemotactic migration, and phagocytic activity [13,14,15,16]. Therefore, the modulation of microglia has been proposed as a therapeutic target for various neurological diseases accompanied by neuroinflammation [17,18,19].

One suggested therapeutic strategy for neuroinflammation is the modulation of ion channels expressed in microglia [20,21,22]. In particular, Ca^2+^ ion signaling is involved in the conversion of microglia from surveillance to activated status [23], and L-type Ca^2+^ channel blockers (CCBs) such as nimodipine and verapamil exhibit neuroprotective effects by inhibiting rodent and human microglial activity [24,25,26]. CCBs derived from another L-type CCB, amlodipine, exert anti-inflammatory effects by inhibiting microglial activation and infiltration [27]. Moreover, beneficial effects of nimodipine, verapamil, and another L-type CCB, isradipine, in neurodegenerative diseases involving neuroinflammation have been reported [28,29,30,31,32]. Interestingly, a recent study demonstrated that the L-type CCB felodipine, which has a long half-life and high blood–brain barrier (BBB) penetration, reduced the number of macrophages in a fructose-induced vascular inflammation model [33]. Furthermore, the administration of felodipine relieved lesions and symptoms in a pulmonary fibrosis model characterized by chronic inflammation [34]. Another study demonstrated that felodipine promotes the removal of toxic proteins (α-synuclein and tau) associated with neurodegenerative diseases by inducing autophagy [35]. However, the effects of felodipine on LPS-induced neuroinflammation have not been investigated. 

In this study, we first evaluated the effects of the L-type CCB felodipine on LPS-stimulated proinflammatory cytokine levels and found that felodipine downregulated LPS-mediated proinflammatory cytokine levels in an L-type Ca^2+^ channel-dependent manner in BV2 microglial cells. In addition, felodipine inhibited TLR4/AKT/STAT3 activation to alter neuroinflammatory responses in BV2 microglial cells. In wild-type mice, felodipine administration (daily for 3 days or 7 days, i.p.) inhibited LPS-induced gliosis and the expression of the proinflammatory cytokines COX-2 and IL-1β under conditions of acute neuroinflammation. Importantly, felodipine reduced short-term memory impairment and hippocampal dendritic spine loss induced by chronic LPS injection in vivo. Moreover, felodipine treatment alleviated chronic neuroinflammation-evoked microgliosis in C57BL/6 mice. Taken together, these results suggest that the L-type CCB felodipine could be repurposed for the treatment of diseases associated with neuroinflammation and cognitive function.

## 2. Results

### 2.1. The L-Type Ca^2+^ Channel Blocker Felodipine Decreases the Expression of LPS-Induced Proinflammatory Cytokines in BV2 Microglial Cells

Felodipine and nitrendipine are L-type Ca^2+^ channel blockers (CCBs) used for the treatment of hypertension. Here, we investigated the effects of felodipine and nitrendipine on LPS-mediated proinflammatory responses in vitro. Before assessing the effects of felodipine or nitrendipine on LPS-induced proinflammatory responses in BV2 microglial cells, which are known to express L-type Ca^2+^ channels [36], we evaluated the cytotoxicity of felodipine or nitrendipine (0.1, 1, 5, 10, 25, and 50 μM) in the cells using an MTT assay. We found that felodipine had no cytotoxicity compared with vehicle at concentrations up to 25 μM but dramatically decreased cell viability at 50 μM (Figure 1a,b). In addition, nitrendipine had no cytotoxicity compared with vehicle at concentrations up to 25 μM (Appendix A).

To determine the dose-dependent effects of felodipine post-treatment on LPS-induced proinflammatory cytokine levels, BV2 microglial cells were treated with 200 ng/mL LPS or PBS for 30 min, followed by felodipine (1, 2.5, or 5 μM) for 5.5 h. Real-time PCR showed that felodipine significantly reduced LPS-mediated proinflammatory cytokine *cox-2* mRNA levels in a dose-dependent manner (Figure 1c). Based on the results of the MTT assays and dose–response experiments, a felodipine concentration of 5 μM was selected for subsequent in vitro experiments.

To further confirm our findings, we performed RT-PCR and found that post-treatment with 5 μM felodipine significantly reduced LPS-evoked *cox-2* mRNA levels but not *il-6*, *il-1β*, and *inos* mRNA levels in BV2 microglial cells (Figure 1d). ELISA analysis verified that felodipine significantly suppressed the LPS-induced increase in secreted COX-2 protein levels in BV2 microglial cells (Figure 1e). In addition, we observed that post-treatment with nitrendipine had no effect on LPS-stimulated proinflammatory cytokine levels (Appendix A).

Next, we assessed the impact of L-type CCB pre-treatment on LPS-evoked proinflammatory cytokine expression. BV2 microglial cells were treated with felodipine (1, 2.5, or 5 μM) or vehicle (1% DMSO) for 30 min, followed by 200 ng/mL LPS or PBS for 5.5 h. Real-time PCR demonstrated that pre-treatment with felodipine significantly diminished LPS-evoked proinflammatory cytokine *cox-2* and *il-1β* mRNA levels in a dose-dependent manner (Figure 1f). To further determine the effects of L-type CCBs on LPS-evoked proinflammatory cytokines, BV2 microglial cells were treated with 5 μM felodipine, 5 μM nitrendipine, or vehicle (1% DMSO) for 30 min and with 200 ng/mL LPS (200 ng/mL) or PBS for 5.5 h. RT-PCR analysis revealed that pre-treatment with felodipine significantly reduced LPS-stimulated *cox-2* and *il-1β* mRNA levels but not *il-6* and *inos* mRNA levels (Figure 1g), whereas nitrendipine pre-treatment significantly reduced LPS-induced *cox-2*, *il-1β,* and *il-6* mRNA levels (Appendix A). ELISA analysis verified that the LPS-induced increases in COX-2 and IL-1β protein levels were significantly decreased by felodipine pre-treatment (Figure 1h). These results indicate that pre-treatment with felodipine or nitrendipine inhibited LPS-induced proinflammatory cytokine expression more effectively than post-treatment.

### 2.2. Felodipine Decreases LPS-Mediated Proinflammatory Cytokine cox-2 and il-1β mRNA Levels in an L-Type Ca^2+^ Channel-Dependent Manner

Since pre-treatment with felodipine or nitrendipine downregulated LPS-induced proinflammatory cytokines, we investigated whether felodipine requires L-type Ca^2+^ channel inhibition to alter the LPS-stimulated proinflammatory response. BV2 microglial cells were transfected with siRNA (60 nM) for the dominant subunit of the L-type Ca^2+^ channel, i.e., *cacna1d*, or scramble (control) siRNA for 24 h, and then, treated sequentially with felodipine (1 or 5 μM) or vehicle (1% DMSO), followed by LPS (200 ng/mL) or PBS. Subsequent real-time PCR showed that transfection with *cacna1d* siRNA successfully reduced *cacna1d* mRNA levels compared with scramble siRNA treatment (Figure 2a,d). In cells in which *cacna1d* was knocked down, treatment with 1 μM felodipine did not significantly alter *cox-2* and *il-1β* mRNA levels compared with LPS treatment (Figure 2b,c), but 5 μM felodipine significantly reduced the LPS-induced increases in *cox-2* and *il-1β* mRNA levels (Figure 2e,f). However, treatment with 5 μM felodipine and LPS significantly increased *cox-2* and *il-1β* mRNA levels in cells in which *cacna1d* was knocked down compared with cells transfected with scramble siRNA and treated with 5 μM felodipine and LPS (Figure 2e,f). Similar results were obtained when 5 μM felodipine was replaced with 5 μM nitrendipine (Appendix A). Based on our findings, we suggest that the suppression of LPS-induced *cox-2* and *il-1β* mRNA expression by 1 μM felodipine is dependent on the L-type Ca^2+^ channel, whereas the effects of 5 μM felodipine are only partially dependent on the L-type Ca^2+^ channel.

### 2.3. Felodipine Downregulated Proinflammatory Cytokine levels via TLR4 Signaling

The binding of LPS to TLR4 activates the AKT and MAPK signaling pathways and transcription factors (e.g., STAT3 and NF-κB) and induces proinflammatory responses [37,38]. To determine whether felodipine and nitrendipine require TLR4 signaling to reduce LPS-mediated proinflammatory cytokine levels, BV2 microglial cells were treated successively with (i) 500 nM TLR4 inhibitor (TAK-242) or vehicle (1% DMSO) for 30 min; (ii) 5 μM felodipine or vehicle (1% DMSO) for 30 min; and (iii) 200 ng/mL LPS or PBS for 5 h. RT-PCR analysis showed that treatment with TAK-242, felodipine, and LPS did not alter LPS-evoked *cox-2* mRNA levels compared with treatment with felodipine and LPS or treatment with TAK-242 and LPS (Figure 2g). Moreover, real-time PCR confirmed that treatment with TAK-242, felodipine, and LPS did not significantly affect LPS-induced *cox-2* mRNA levels compared with treatment with felodipine and LPS or treatment with TAK-242 and LPS (Figure 2h). Similar results were obtained when felodipine was replaced with nitrendipine (Appendix A). These results suggest that there is an inhibitory effect of felodipine or nitrendipine on LPS-induced *cox-2* expression through TLR4 signaling.

### 2.4. Felodipine Decreases LPS-Induced Proinflammatory Cytokine Levels by Decreasing AKT and STAT3 Phosphorylation in BV2 Microglial Cells

To test whether the L-type CCBs felodipine and nitrendipine affect LPS-mediated AKT and/or MAPK signaling in vitro, BV2 microglial cells were treated with 5 μM felodipine or vehicle (1% DMSO) for 45 min, followed by 200 ng/mL LPS or PBS for 45 min, and Western blotting with anti-p-AKT^S473^ and anti-AKT antibodies. The Western blot results showed that felodipine significantly downregulated LPS-mediated p-AKT^S473^ levels in BV2 microglial cells (Figure 3a). Further verifying these findings, immunocytochemistry with an anti-p-AKT^S473^ antibody showed that felodipine significantly suppressed LPS-induced AKT^S473^ phosphorylation (Figure 3b). Similar results were obtained when felodipine was replaced with nitrendipine (Appendix A). However, felodipine did not alter LPS-stimulated p-AKT^Y308^ levels in BV2 microglial cells (Appendix A). Moreover, felodipine did not affect LPS-mediated ERK and P38 phosphorylation compared with LPS treatment (Appendix A).

Given the effect of felodipine on LPS-induced AKT phosphorylation, we examined whether felodipine regulates LPS-evoked proinflammatory responses in an AKT-dependent manner. In this experiment, BV2 microglial cells were treated successively with (i) 10 μM AKT inhibitor (MK-2206) or vehicle (1% DMSO) for 30 min; (ii) 5 μM felodipine or vehicle (1% DMSO) for 30 min; and (iii) 200 ng/mL LPS or PBS for 5 h. Subsequent RT-PCR showed that treatment with MK-2206, felodipine, and LPS did not significantly alter LPS-mediated *cox-2* mRNA levels compared with treatment with felodipine and LPS or with MK-2206 and LPS (Figure 3c). By using real-time PCR, we further confirmed that treatment with MK-2206, felodipine, and LPS did not significantly alter LPS-stimulated *cox-2* mRNA levels compared with treatment with felodipine and LPS or with MK-2206 and LPS (Figure 3d). These results indicate that felodipine downregulates LPS-stimulated proinflammatory cytokine levels by inhibiting AKT activation in BV2 microglial cells.

Since felodipine and nitrendipine modulate AKT signaling, which is downstream of the L-type Ca^2+^ channel and TLR4 signaling, we examined the effects of these L-type CCBs on LPS-stimulated nuclear STAT3 and/or NF-κB phosphorylation in BV2 microglial cells. The BV2 microglial cells were exposed to 5 μM felodipine or vehicle (1% DMSO) for 30 min, followed by 200 ng/mL LPS or PBS for 5.5 h, and then, subjected to nuclear fractionation. We found that felodipine pre-treatment significantly reduced LPS-mediated nuclear p-STAT3^S727^ levels in BV2 microglial cells (Figure 3e). To verify these findings, we performed immunocytochemistry of the treated cells, which showed that felodipine pre-treatment significantly diminished the LPS-induced increase in STAT3 phosphorylation (Figure 3f). We also found that nitrendipine had similar effects on LPS-evoked nuclear STAT3 phosphorylation in BV2 microglial cells (Appendix A). By contrast, felodipine did not alter the LPS-induced increase in nuclear NF-κB phosphorylation (Appendix A). These results suggest that felodipine and nitrendipine modulate LPS-induced STAT3 activation, leading to the attenuation of LPS-induced proinflammatory cytokine release in BV2 microglial cells.

### 2.5. Daily Intraperitoneal Administration of Felodipine for 3 Days Alleviates LPS-Induced Microgliosis in the Brain in C57BL/6 Mice

Although both felodipine and nitrendipine downregulated the neuroinflammatory response in vitro, we chose felodipine for subsequent in vivo experiments due to its longer half-life and greater BBB penetration compared with nitrendipine [34]. To induce acute and chronic neuroinflammation in vivo, we injected 3-month-old C57BL/6 mice with LPS, which triggers the TLR4-mediated inflammatory signaling pathway and induces responses mimicking neuroinflammation-related diseases [39,40,41]. Based on the literature, we selected 5 mg/kg felodipine and intraperitoneal (i.p.) administration [35]. 

First, we examined whether felodipine can affect LPS-evoked neuroinflammatory response in vivo, whereby wild-type mice were injected with felodipine (5 mg/kg, i.p.) or vehicle (5% Tween-20 + 5% PEG300 in saline) daily for 3 days. On day 3, LPS (10 mg/kg, i.p.) or PBS was intravenously (i.v.) injected to induce acute neuroinflammation (Figure 4a). After 8 h, the mouse brains were sacrificed, and the brain tissues were immunostained with an anti-Iba-1 antibody. The designation of the regions of interest (ROI) in the immunostained hippocampus is shown in Figure 4b. The administration of felodipine daily for 3 days significantly reduced the LPS-induced increases in Iba-1 fluorescence intensity, Iba-1-positive cells and the fractionation of the Iba-1-positive area in the cortex (Figure 4c,d). In the hippocampal CA1 and DG regions, the administration of felodipine daily for 3 days significantly decreased the LPS-induced increase in the number of Iba-1-positive cells but not Iba-1 fluorescence intensity or the fractionation of the Iba-1-positive area (Figure 4c,d). Moreover, daily administration of felodipine for 3 days did not affect the LPS-induced increases in Iba-1 fluorescence intensity, the number of Iba-1-positive cells, or the Iba-1 % of the area fraction in the hippocampal CA3 region (Figure 4c,d). These results suggest that daily administration of felodipine for 3 days selectively regulates LPS-induced microglial activation and morphology in regions of the brain in C57BL/6 mice.

### 2.6. Daily Intraperitoneal Administration of Felodipine for 3 Days Downregulates LPS-Induced COX-2 and IL-1β Levels in the Brain in C57BL/6 Mice

Since felodipine reduced LPS-stimulated microgliosis in the brain as shown in Figure 4, we examined whether felodipine alters LPS-induced proinflammatory cytokine levels in vivo. C57BL/6 mice were injected with felodipine (5 mg/kg, i.p.) or vehicle (5% Tween-20+ 5% PEG300 in saline) daily for 3 days, followed by LPS (10 mg/kg) or PBS on day 3 as described above, and immunofluorescence staining of brain sections was performed with anti-COX-2 or anti-IL-1β antibodies. Interestingly, daily felodipine administration for 3 days significantly reduced LPS-mediated COX-2 and IL-1β levels in the cortex and hippocampus in C57BL/6 mice (Figure 5a–d). These results indicate that daily treatment with felodipine for 3 days inhibits LPS-stimulated proinflammatory cytokine COX-2 and IL-1β levels in the brain, which may contribute to the effects of felodipine on LPS-stimulated microgliosis.

### 2.7. Daily Intraperitoneal Administration of Felodipine for 7 Days Alleviates LPS-Induced Microgliosis and Astrogliosis in the Brain in C57BL/6 Mice

Although 3 days of daily i.p. injection of felodipine significantly reduced LPS-stimulated microgliosis in the cortex, LPS-induced microgliosis in the hippocampus was only slightly suppressed (Figure 4). To further determine whether longer treatment with felodipine differentially affects LPS-induced gliosis in vivo, C57BL/6 mice were injected with felodipine (5 mg/kg, i.p.) or vehicle (5% Tween-20 + 5% PEG300 in saline) daily for 7 days. On the 7th day, LPS (10 mg/kg) or PBS was intraperitoneally (i.p.) injected after the last felodipine administration. The mice were sacrificed 8 h after the LPS or PBS injection, and anti-Iba-1 and anti-GFAP antibodies were used for immunofluorescence staining of the brain sections. Importantly, daily administration of felodipine for 7 consecutive days significantly attenuated the Iba-1 fluorescence intensity in the cortex and hippocampal CA1 and DG regions (Figure 6a,b).

The Iba-1-positive cells and the fractionation of the Iba-1-positive area were significantly reduced by felodipine administration (daily for 7 days) in the cortex and hippocampal CA1, DG, and CA3 regions after LPS injection (Figure 6a,b). In addition, 7 consecutive days of felodipine injection significantly suppressed the LPS-induced increase in GFAP fluorescence intensity in the cortex but not the hippocampus (Figure 6c,d). Daily felodipine injection for 7 days reduced the LPS-induced increase in the GFAP-positive cell number in the hippocampal DG and CA3 regions but not in the cortex and hippocampal CA1 region (Figure 6c,d). Moreover, daily administration of felodipine for 7 days significantly decreased the LPS-induced increase in the fractionation of the GFAP-positive area in the cortex and hippocampal CA1, DG, and CA3 regions (Figure 6c,d). These data demonstrate that a longer period of treatment with felodipine is more effective than a short regimen for alleviating micro/astrogliosis in vivo.

### 2.8. Intraperitoneal Administration of Felodipine Daily for 7 Days Downregulates LPS-Stimulated COX-2 and IL-1β Levels in the Brain in C57BL/6 Mice

To investigate whether a longer period of felodipine treatment modulates LPS-mediated proinflammatory cytokine levels, 3-month-old C57BL/6 mice were injected daily with felodipine (5 mg/kg, i.p.) or vehicle (5% Tween-20 + 5% PEG300 in saline) for 7 days, followed by injection of LPS (10 mg/kg, i.p.) or PBS on day 7, 30 min after the last felodipine injection. Immunostaining of brain sections with anti-COX-2 and anti-IL-1β antibodies showed that treatment with felodipine daily for 7 days significantly suppressed the LPS-mediated elevation of COX-2 and IL-1β fluorescence intensity in the cortex and hippocampus (Figure 7a–d). These results suggest that longer felodipine treatment diminishes LPS-evoked proinflammatory cytokine COX-2 and IL-1β levels in C57BL/6 mice.

### 2.9. Intraperitoneal Administration of Felodipine Daily for 9 Days Regulates LPS-Induced Spatial Memory Impairment, Dendritic Spine Formation, and Microgliosis in C57BL/6 Mice

Recent studies have reported that chronic LPS administration promotes neuroinflammation-mediated cognitive disruption [42,43,44,45] and that low-dose (250 μg/mL) chronic LPS administration induces motor and short-term working memory impairment in wild-type mice [46]. Based on these observations, we used a low dose of LPS to evaluate the effects of the L-type CCB felodipine on chronic neuroinflammation-mediated learning and memory impairments. Specifically, 3-month-old C57BL/6 male mice were injected with felodipine (5 mg/kg, i.p.) or vehicle (5% Tween-20 + 5% PEG300 in saline) daily for 9 days. On each of the 9 days, the felodipine (5 mg/kg, i.p.) or vehicle injection was followed 30 min later by injection of 250 μg/mL LPS or PBS.

On day 7 of the treatment regimen, Y-maze tests were conducted to investigate whether felodipine modulates the LPS-induced impairment of short-term spatial working memory. On days 8–9 of the treatment regimen, novel object recognition (NOR) training and testing were performed to evaluate the effect of felodipine on the LPS-mediated impairment of long-term recognition memory. Importantly, felodipine administration daily for 9 days significantly restored the LPS-induced decrease in spontaneous alternations but did not affect the total number of entries (Figure 8a). By contrast, treatment with felodipine daily for 9 days did not alter the novel object preference of LPS-treated C57BL/6 mice (Figure 8b).

Short-term spatial memory is strongly correlated with dendritic spinogenesis in the hippocampus [47,48]. Since felodipine reduced chronic neuroinflammation-mediated short-term spatial memory impairment in LPS-treated C57BL/6 mice, we investigated whether felodipine alters dendritic spine formation in LPS-treated C57BL/6 mice. Golgi staining after the behavior tests showed that felodipine administration daily for 9 days significantly reduced the chronic neuroinflammation-induced decrease in the basal shaft (BS) dendritic spine number in the hippocampus (Figure 8c,d). However, daily felodipine administration for 9 days did not affect the apical oblique (AO) spine number in the hippocampus in LPS-treated C57BL/6 mice (Figure 8c,d). These data indicate that felodipine contributes to ameliorating chronic neuroinflammation-induced short-term memory impairment by modulating hippocampal dendritic spine formation.

We further examined whether the felodipine treatment regimen (daily for 9 days, 5 mg/kg, i.p.) affected LPS-induced neuroinflammation. We found that the regimen significantly reduced the LPS-induced increases in Iba-1 fluorescence intensity, the number of Iba-1-positive cells, and the fractionation of the Iba-1-positive area in the cortex and hippocampus (Figure 8e,f). These results suggest that felodipine alleviates chronic neuroinflammation-induced microgliosis in the brain in C57BL/6 mice.

## 3. Discussion

Various neurological diseases, including neurodegenerative diseases, are accompanied by neuroinflammation. Therefore, the regulation of microglia, which play a pivotal role in neuroinflammation, has been regarded as a therapeutic target [18,19,20]. In this study, we demonstrated that the L-type CCB felodipine significantly suppressed LPS-induced proinflammatory cytokine release via L-type Ca^2+^ channel/TLR4/AKT signaling pathways in BV2 microglial cells. In addition, felodipine inhibited LPS-stimulated nuclear STAT3 activation in vitro. In in vivo experiments, felodipine administration daily for 7 days reduced LPS-evoked gliosis in the brain more effectively than felodipine administration daily for 3 days, although both treatments reduced proinflammatory cytokine levels. Moreover, felodipine restored chronic LPS-mediated impairment of short-term spatial memory by promoting hippocampal dendritic spine formation (Figure 9). These results may support the repurposing of the L-type CCB felodipine for the treatment of neuroinflammation and impaired cognitive function. 

Accumulating evidence indicates that Ca^2+^ dynamics in microglia play an important role in the regulation of neuroinflammation in the CNS [49]. Microglia are known as innate immune cells of the CNS and maintain homeostasis under normal conditions [50]. They can transiently respond to injury or pathogens by shifting their morphology, resulting in the release of cytokines and ROS [7]. LPS stimulates the transformation of microglia from the resting state to the activated state by regulating the influx of intracellular Ca^2+^ [51]. LPS-primed microglial Ca^2+^ transients have been shown to increase IL-1β and COX-2 levels [52]. Another L-type CCB, nimodipine, inhibits LPS-induced proinflammatory cytokine COX-2 and NO production in BV2 microglial cells [53]. Furthermore, Ca^2+^ contributes to IL-1β maturation by regulating the NLRP3 inflammasome together with cAMP, and Ca^2+^ influx activates the Ca^2+^-dependent pathway to upregulate COX-2 expression in microglia [54,55]. Importantly, LPS-activated microglia display elevated levels of basal Ca^2+^ and reduced Ca^2+^ channel-dependent stimulation, indicating that activated microglia can ignore additional stimuli [51]. However, the effects of other L-type CCBs on LPS-induced proinflammatory responses and the molecular mechanisms of action of this class of drugs have received little attention. Thus, we examined the effects of the first-generation L-type CCB nitrendipine and the second-generation L-type CCB felodipine on LPS-induced proinflammatory responses.

We found that pre-treatment with felodipine or nitrendipine was more effective than post-treatment in reducing proinflammatory cytokine levels (Figure 1 and Appendix A). Taken together with the literature, our findings indicate that treating BV2 microglial cells with felodipine or nitrendipine before LPS exposure may inhibit the release of proinflammatory cytokines more effectively than treatment after exposure. 

Felodipine blocks the L-type Ca^2+^ channel directly in vivo and in vitro [56,57]. Thus, we investigated the involvement of the L-type Ca^2+^ channel in the effects of felodipine on LPS-induced proinflammatory cytokine levels in vitro by knocking down the expression of the dominant subunit of the L-type Ca^2+^ channel, *cacna1d*. We found that 1 μM felodipine downregulated LPS-mediated proinflammatory responses in an L-type Ca^2+^ channel-dependent manner, whereas the effect of 5 μM felodipine on LPS-stimulated proinflammatory responses was only partially dependent on the L-type Ca^2+^ channel (Figure 2). These data indicate that a blockade of the L-type Ca^2+^ channel by 1 μM felodipine leads to the inhibition of intracellular Ca^2+^ influx and, consequently, the suppression of LPS-mediated proinflammatory cytokine expression. Why does a high dose of felodipine (5 μM) differentially regulate LPS-stimulated proinflammatory responses? It is possible that 5 μM felodipine completely blocks the L-type Ca^2+^ channel, leading to further inhibition of off-targets of felodipine (e.g., phosphodiesterase (PDE)). A few studies have reported that felodipine binds and inhibits calcium-binding proteins such as calmodulin, calmodulin-dependent cyclic nucleotide phosphodiesterase (PDE1), and the voltage-gated T-type calcium channel [58,59]. Interestingly, PDE inhibitors can suppress microglial activation and proinflammatory cytokines in vivo and in vitro [60,61]. In addition, it is reported that antisense oligos (ASOs) significantly reduce on-target (BACH1) and off-target (SRRM2, TLE3, USP9X) mRNAs in a dose- and time-dependent manner in vitro, indicating that the drug can inhibit off-targets in a dose-dependent manner [62]. In the context of these observations in the literature, our findings suggest that felodipine differentially suppresses LPS-induced neuroinflammation by inhibiting the L-type Ca^2+^ channel (on-target) and/or other proteins (i.e., off-target, such as PDE) in a dose-dependent manner in vitro and in vivo.

Increased Ca^2+^ influx facilitates COX-2 expression in various cell types, including microglia [55,63,64]. In addition, increased cytosolic Ca^2+^ is essential for LPS-mediated macrophage activation and TLR4 membrane trafficking [65]. These findings suggest that a blockade of Ca^2+^ channels through CCBs may not only reduce LPS-mediated microglial activation but may also contribute to the modulation of COX-2 expression through TLR4 signaling. Another CCB, nicardipine, significantly inhibits proinflammatory cytokine IL-1β release by suppressing LPS-induced TLR4 expression [66]. We found that felodipine and nitrendipine did not significantly alter LPS-mediated proinflammatory cytokine *cox-2* mRNA levels through treatment with TLR4 inhibitor in BV2 microglial cells (Figure 2 and Appendix A), indicating that felodipine and nitrendipine modulates TLR4 signaling to ameliorate LPS-induced proinflammatory cytokine levels in vitro.

LPS stimulation of TLR4 triggers the phosphorylation of downstream signaling molecules such as AKT and MAPKs (e.g., ERK1/2 and P38) [67,68,69]. Several L-type CCBs have been shown to inhibit the phosphorylation of AKT and other MAPKs in microglia and other cells [53,70]. For instance, the Ca^2+^ channel blocker nicardipine (10 μM) decreases the LPS-induced increase in 17hosphor-AKT levels in BV2 microglial cells [53]. Another L-type CCB, nifedipine, significantly reduces AKT phosphorylation in vascular smooth muscle cells [70]. Here, we found that felodipine and nitrendipine significantly inhibited LPS-mediated p-AKT^S473^ levels in BV2 microglial cells (Figure 3). However, felodipine did not affect LPS-induced p-AKT^Y308^ levels (Appendix A), suggesting that felodipine specifically inhibits AKT phosphorylation at serine 473 in vitro. In addition, we found that felodipine is necessary for AKT inhibition to downregulate LPS-mediated proinflammatory responses in BV2 microglial cells (Figure 3). Overall, these data suggest that felodipine selectively inhibits AKT phosphorylation but not ERK/P38 signaling to diminish LPS-evoked proinflammatory responses in BV2 microglial cells.

The TLR4-mediated phosphorylation cascade activates the transcription factors NF-κB, and STAT3, which directly regulate the transcription of proinflammatory cytokines [68,71,72,73]. In addition, Ca^2+^ influx and AKT are implicated in the activation of NF-κB and STAT3 [74,75], and some L-type CCBs have been reported to inhibit these transcription factors [76,77,78]. For example, both nimodipine and verapamil significantly suppress IFN-γ and LPS-stimulated STAT3 phosphorylation in human-derived microglial cells [24]. We found that felodipine treatment significantly inhibited nuclear STAT3 phosphorylation in LPS-treated BV2 microglial cells (Figure 3). However, felodipine treatment did not alter LPS-induced nuclear p-NF-kB levels in vitro (Appendix A). Overall, our findings and the literature suggest that felodipine alters LPS-mediated proinflammatory responses via STAT3 inhibition without NF-κB involvement in BV2 microglial cells. However, a recent study found that felodipine (5 mg/kg, daily injection for 6 weeks) downregulated NF-κB in a vascular inflammation model [33]. Another study demonstrated that felodipine treatment (5 mg/kg/day, daily injection for 12 weeks) decreased high-cholesterol-diet-induced vascular inflammation factors and p-NF-κB levels in a rat model of atherosclerosis [79]. Thus, it is possible that longer felodipine treatment regimens (24 or 48 h for in vitro experiments, daily injection for 6 or 12 weeks for in vivo experiments) would inhibit LPS-stimulated nuclear NF-κB phosphorylation in vitro and/or in vivo. In a future study, we will elucidate the detailed anti-inflammatory mechanism of felodipine by clarifying its effects on STAT3/NF-κB regulation.

A previous study reported that stimulation of L-type Ca^2+^ channels increased the activation of LPS-induced reactive astrocytes in vitro [80]. In addition, the L-type-specific CCB verapamil significantly diminishes the LPS-induced increase in proinflammatory cytokine release in primary cortical astrocytes [81], and the L-type CCBs nimodipine and verapamil suppress IFN-γ levels in human astrocytes [24]. In a cuprizone-induced neuroinflammation-associated myelin injury mouse model, astrocyte-specific deletion of Cav1.2 significantly reduced astrocyte activation in the brain [82]. Interestingly, we found that daily felodipine treatment for 7 days reduced LPS-mediated microglial activation and proinflammatory cytokine COX-2 and IL-1β levels more effectively than a 3-day treatment regimen in wild-type mice (Figure 4, Figure 5, Figure 6 and Figure 7). In addition, we observed that daily felodipine treatment for 7 days had smaller effects on LPS-stimulated astrogliosis than on LPS-induced microgliosis in wild-type mice (Figure 6). Thus, a treatment period of 7 days may not be sufficient to observe significant effects of felodipine on LPS-mediated astrogliosis in vivo. In future work, we will use longer treatment periods (e.g., daily for 1 month) to assess the effects of felodipine on LPS-induced astrogliosis and LPS-mediated proinflammatory cytokine release. Moreover, we will probe the molecular mechanism of action of felodipine in vivo using an AAV siRNA virus system. 

In the present study, we focused on the effects of felodipine on LPS-induced neuroinflammatory responses in the brain, but it is possible that felodipine can also affect peripheral inflammation. Indeed, a recent study demonstrated that felodipine treatment effectively inhibits peripheral inflammation in vivo [33]. Specifically, repeated administration of 5 mg/kg felodipine significantly reduced high-fructose-diet-induced activation of macrophages in Wistar rats [33]. However, few studies have examined the effects and molecular mechanisms of action of felodipine on inflammatory responses in vivo. In subsequent work, we will address the effects of felodipine on neuroinflammatory and peripheral inflammatory responses and identify the molecular targets involved in peripheral inflammation versus neuroinflammation in the brain.

Microglia and astrocytes participate in cross-talk with each other, and the effects of felodipine may reflect its combined influence on both cell types. This study does not provide direct evidence that felodipine influences the interaction between microglia and astrocytes, but reducing the levels of the proinflammatory cytokines COX-2 and IL-1β is likely to impact both cell types in vivo (Figure 6 and Figure 7). Astrocytes, microglia, and oligodendrocytes are essential cells for controlling neuronal homeostasis in the CNS [83]. In particular, communication between astrocytes and microglia modulates neuroinflammation via the secretion of several cytokines and inflammatory mediators [84]. A previous study demonstrated that LPS-induced Ca^2+^ influx activated microglia and the release of IL-1α and TNF-α, resulting in reactive astrocyte morphology in the CNS [85]. In this study, we demonstrated that daily injection of felodipine for 7 successive days significantly reduced reactive microglia, astrocytes, and proinflammatory cytokine levels in vivo (Figure 6 and Figure 7). According to the literature and our findings, felodipine may initially inhibit microglial activation by blocking the L-type Ca^2+^ channel and/or off-target effects, and subsequently suppress astrogliosis by interacting with microglia. 

In exploring the preventive and/or curative effects of felodipine on LPS-induced neuroinflammatory responses in vitro, we observed that pre-treatment with felodipine (preventive) diminished LPS-induced proinflammatory responses more effectively than post-treatment (curative). Thus, we further investigated whether pre-treatment with felodipine up- or downregulates LPS-stimulated neuroinflammation in vivo (Figure 4, Figure 5, Figure 6 and Figure 7). A previous study reported that post-treatment with the L-type CCB Zinc20267861 significantly reduced microglial activation in a mouse model of suture-induced inflammation [27]. Another study demonstrated that post-treatment with the L-type CCB nimodipine significantly suppresses LPS-induced proinflammatory cytokine levels and Iba-1-positive cells in the hippocampus in rats [86]. These observations and our findings suggest that both pre- and post-treatment with L-type CCBs modulate neuroinflammatory responses in vivo. We will further address the effects of felodipine post-treatment on LPS-mediated neuroinflammatory responses in vivo in future work. 

Several studies have reported a close link between neuroinflammation and memory impairment, including a correlation between LPS-induced neuroinflammation and cognitive function in vivo [42]. For example, peripheral IL-1β administration and intra-hippocampal IL-1β infusion-induced chronic neuroinflammation impair short-term spatial and contextual memory in mice [87]. Supporting in vivo evidence shows that LPS treatment (4 mg/kg, daily i.p. injection for 14 successive days) suppresses synaptic protein expression and micro-/astroglial activation in C57BL/6 mice, indicating that neuroinflammation is associated with cognitive function [88]. Interestingly, several recent studies have shown that felodipine modulates microglial Ca^2+^ signals linked to neuronal activity and synaptic development [89,90]. Beyond neuroinflammation, the disruption of Ca^2+^ signaling is also associated with synaptic dysfunction and memory loss [91,92]. Moreover, chronic neuroinflammation leads to a deficiency in L-type Ca^2+^ channel-dependent long-term potentiation (LTP) in hippocampal CA1 neurons, resulting in spatial memory impairment [93,94]. These previous studies suggest that L-type Ca^2+^ channels are required to sustain homeostasis in learning and memory, but chronic neuroinflammation-induced impairments in learning and memory have rarely been explored. Here, we found that felodipine reduced the decrease in spontaneous alternations in the Y-maze test evoked by repeated LPS injection and an increased hippocampal basal shaft (BS) dendritic spine number (Figure 8). Taken together, the literature and our findings suggest that the L-type CCB felodipine modulates hippocampal-dependent spatial memory via the upregulation of dendritic spine formation in this mouse model of chronic neuroinflammation. Of course, other explanations are possible. For instance, felodipine may improve spatial memory and the dendritic spine number via Ca^2+^ conductance and L-type Ca^2+^ channel subunit activity/expression in the hippocampal CA1 region. Therefore, future work will examine additional underlying mechanisms by which felodipine may improve synaptic/cognitive function, including region-specific synaptic plasticity.

## 4. Materials and Methods

### 4.1. Ethics Statement

All experiments were approved by the institutional biosafety committee (IBC) and performed in accordance with approved animal protocols of the Korea Brain Research Institute (KBRI, approval no. IACUC-20-00024 and IACUC-21-00021).

### 4.2. L-Type Ca^2+^ Blockers Felodipine and Nitrendipine

Felodipine (Cat. No. F0814; Tokyo Chemical Industry, Tokyo, Japan) and nitrendipine (Cat. No. 17549; Cayman Chemical Company, Ann Arbor, MI, USA) were dissolved in dimethyl sulfoxide (1% DMSO) for in vitro experiments or 5% Tween-20 + 5% PEG300 in saline for in vivo experiments.

### 4.3. BV2 Microglial Cell Culture

BV2 microglial cells were kindly gifted by Dr. Kyungho Seok of Kyungpook National University. The cells were maintained in high-glucose Dulbecco’s modified Eagle’s medium (DMEM; Cat. No. SH30243.01; HyClone Laboratories, Logan, UT, USA) in the presence of 1% penicillin–streptomycin (Cat. No. 30-002-CI; Corning, NY, USA) and 5% fetal bovine serum (Cat. No. SH30084.03HI; HyClone Laboratories). The cells were seeded at a density of 4 × 10^4^ cells/mL for cytotoxicity tests; 2.5 × 10^5^ cells/mL for RT-PCR, real-time PCR, Western blot, and ELISA; and 4 × 10^5^ cells/mL for immunocytochemistry. After incubation overnight, the cells were starved in serum-free DMEM for 1 h, and the treatments for the in vitro experiments were administered.

### 4.4. Cytotoxicity Test (MTT Assay)

3-(4,5-Dimethylthiazol-2-yl)-2,5-diphenyltetrazolium bromide (MTT; Cat. No. 0793; VWR Chemicals, Radnor, PA, USA) was used to evaluate the cytotoxicity of the L-type CCBs in BV2 microglial cells. The cells were cultured with 0.1, 1, 5, 10, 25, or 50 μM felodipine or nitrendipine for 24 h, and then, treated with 0.5 mg/mL MTT for 3 h. After incubation with DMSO (Cat. No. D8418; Sigma-Aldrich, St. Louis, MO, USA) for 10 min to dissolve the formazan product, the absorbance of the cell lysate was measured at 570 nm in a SPECTROstar Nano microplate reader (BMG LABTECH, Ortenberg, Germany).

### 4.5. Reverse Transcription–Polymerase Chain Reaction (RT-PCR)

To investigate the effects of the L-type CCBs on the expression of LPS-induced proinflammatory cytokine genes in vitro, BV2 microglial cells were treated first with felodipine (1, 2.5, or 5 μM), nitrendipine (5 μM), or vehicle (1% DMSO) for 30 min, and then, with 200 ng/mL LPS (Sigma, Cat. No. L2630, *Escherichia coli*) or PBS for 5.5 h. The role of TLR4 signaling in the effects of L-type CCBs on LPS-mediated proinflammatory cytokine levels was investigated by pre-treating cells with the TLR4 inhibitor TAK-242 (Cat. No. 614316; Sigma-Aldrich) for 30 min before sequential treatment with 5 μM L-type CCBs or vehicle for 30 min, and with 200 ng/mL LPS or PBS for 5 h. The role of AKT signaling in the effects of felodipine on LPS-mediated proinflammatory cytokine levels was explored by pre-treating cells with the AKT inhibitor MK-2206 (Cat. No. S1078; Selleckchem, Houston, TX, USA) for 30 min before sequential treatment with 5 μM felodipine or vehicle for 30 min, and with 200 ng/mL LPS or PBS for 5 h. RNA was extracted from the treated cells using QIAzol Lysis Reagent (Cat. No. 79306; QIAGEN, Hilden, Germany). cDNA synthesis and RT-PCR were performed according to the manufacturer’s protocols using SuPrimeScript RT premix (Cat. No. SR-5000; GENET BIO, Daejeon, Korea) and Prime Taq premix (Cat. No. G-3000; GENET BIO), respectively. The primer sequences were as follows: *il-1β*: sense 5′-AGC TGG AGA GTG TGG ATC CC-3′, antisense 5′-CCT GTC TTG GCC GAG GAC TA-3*′*; *il-6*: sense 5′-CCA CTT CAC AAG TCG GAG GC-3′, antisense 5′-GGA GAG CAT TGG AAA TTG GGG T-3*′*; *cox-2*: sense 5′-GCC AGC AAA GCC TAG AGC-3′, antisense 5*′*-GCC TTC TGC AGT CCA GGT TC-3*′*; *inos*: sense 5*′*-CCG GCA AAC CCA AGG TCT AC-3′, antisense 5′-GCA TTT CGC TGT CTC CCC AA-3′; *gapdh*: sense 5*′*-CAG GAG CGA GAC CCC ACT AA-3′, antisense 5′-ATC ACG CCA CAG CTT TCC AG-3′.

### 4.6. Cacna1d siRNA Transfection

The dependence of the effects of felodipine or nitrendipine on LPS-evoked inflammatory responses on L-type Ca^2+^ channels was assessed by knocking down the L-type Ca^2+^ channel alpha subunit (*cacna1d*) in BV2 microglial cells. *cacna1d* or scramble (control) siRNA (Dharmacon, Rafayett, CO, USA) was diluted in Opti-MEM medium (Thermo Scientific, Waltham, MA, USA) and incubated with 1 µL of Lipofectamine*^®^* RNAiMAX reagent (Thermo Scientific, Waltham, MA, USA) for 30 min. The final siRNA concentration was 60 nM. The siRNA mixture was then added to cells (2.5 × 10^5^ cells/well) in a 24-well cell culture plate. After 24 h, 1 or 5 μM L-type CCB or 1% DMSO (vehicle) was added, and 30 min later, 200 ng/mL LPS or PBS was added. Finally, real-time PCR was performed 6 h after L-type CCB or vehicle addition to measure the mRNA levels of *cacna1d* and the proinflammatory cytokines *cox-2* and *il-1β*.

### 4.7. Real-Time PCR

A previously reported method for real-time PCR was used to evaluate the mRNA levels of *cacna1d*, *cox-2,* and *il-1β* in BV2 microglial cells [95,96]. In brief, the Superscript cDNA Premix Kit II (GeNet Bio, Daejeon, Korea) was used to prepare cDNA, which was used in 40-cycle real-time PCR using Fast SYBR Green Master Mix (Thermo Fisher Scientific, San Jose, CA, USA) and a QuantStudio™ 5 system (Thermo Fisher Scientific). Normalization was performed using the cycle threshold (Ct) value for GAPDH. The fold change in the mRNA level relative to the control (vehicle) was calculated. The primer sequences were as follows: *il-1β*: sense 5′-TTG ACG GAC CCC AAA AGA TG-3′, antisense 5′-AGG ACA GCC CAG GTC AAA G-3′; *cox-2*: sense 5′-CCA CTT CAA GGG AGT CTG GA -3′, antisense 5′-AGT CAT CTG CTA CGG GAG GA-3′; *cacna1d*: sense 5′-GTG GAA GTG TGA GCG GAT TGA C-3′, antisense 5′- TCG CTT GAA CCA GGT GCT GGA A-3′; *gapdh*: sense 5′-TGG GCT ACA CTG AGG ACC ACT-3′, antisense 5′-GGG AGT GTC TGT TGA AGT CG-3′.

### 4.8. Enzyme-Linked Immunosorbent Assay (ELISA)

The levels of IL-1β and COX-2 in the media of BV2 microglial cells were measured using ELISA kits (IL-1β: Cat. No. 88-7013-88, Invitrogen, Waltham, MA, USA; COX-2: Cat. No. DYC4198-5, R&D Systems, Minneapolis, MN, USA) as previously described [46]. The conditioned medium used in the ELISA was harvested from cells and treated successively with 5 μM felodipine or vehicle for 30 min and with 200 ng/mL LPS or PBS for 5.5 h. The ELISA was conducted according to the provided protocol.

### 4.9. Nuclear Fractionation

To assess p-STAT3 levels in the nucleus, BV2 microglial cells were treated with felodipine or nitrendipine as described above for ELISA. The cells were then lysed for 5 min in 120 μL of cytosolic fractionation buffer (10 mM HEPES pH 8.0, 1.5 mM MgCl_2_, 10 mM KCl, 0.5 mM DTT, 300 mM sucrose, 0.5 mM PMSF, and 0.1% NP-40) and centrifuged at 10,000 rpm and 4 °C for 1 min. The supernatant (cytosolic fraction) was removed, and the pellet was resuspended in nuclear fractionation buffer (80 μL; 10 mM HEPES pH 8.0, 100 mM KCl, 100 mM NaCl, 0.2 mM EDTA, 0.5 mM DTT, 0.5 mM PMSF, and 20% glycerol) for 15 min on ice. After centrifugation at 10,000 rpm and 4 °C for 15 min, the nuclear fraction was used to assess nuclear p-STAT3 levels via Western blot; proliferating cell nuclear antigen (PCNA, Cat. No. SC-56; Santa-Cruz, Dallas, TX, USA) was used as the control.

### 4.10. Western Blot

PRO-PREP™ Protein Extraction Solution (Cat. No. 17081; iNtRON Biotechnology, Seongnam-si, Korea) containing PhosSTOP™ phosphatase inhibitor (Cat. No. 04 906845001; Sigma-Aldrich) was used to harvest proteins from BV2 microglial cells. Proteins were quantified by performing a DC Protein Assay (Cat. No. 500-0116; Bio-Rad, Hercules, CA, USA), and 20 μg of protein in Laemmli Sample Buffer (Cat. No. 161-0747; Bio-Rad) was loaded into each well of an 8% acrylamide gel. Proteins were separated by electrophoresis for 1 h at 100V and transferred to a polyvinylidene difluoride membrane, which was immunoblotted with antibodies against p-AKT^S473^ (Cat. No. 9271; Cell Signaling, Danvers, MA, USA), p-AKT^Y308^ (Cat. No. 9275; Cell Signaling), AKT (Cat. No. 9272S; Cell Signaling), p-ERK^T202/Y204^ (Cat. No. 9101; Cell Signaling), ERK (Cat. No. 9102S; Cell Signaling), p-P38^T180/Y182^ (Cat. No. 9211; Cell Signaling), P38 (Cat. No. 9212; Cell Signaling), p-STAT3^S727^ (Cat. No. ab86430; Abcam, Cambridge, UK), and STAT3 (Cat. No. 9139; Cell Signaling). The immunoblots were visualized using Lumigen ECL Ultra (Cat. No. TMA-100; Lumigen, Southfield, MI, USA).

### 4.11. Immunocytochemistry (ICC)

BV2 microglial cells that had been seeded and cultured on pre-cleaned cover glass were fixed in 4% PFA for 10 min. The fixed cells were then incubated with antibodies against p-AKT^S473^ (Cat. No. 9271; Cell Signaling), p-AKT^Y308^ (Cat. No. 9275; Cell Signaling), p-NF-κB^S536^ (Cat. No. 3033S; Cell Signaling), or p-STAT3^S727^ (Cat. No. ab86430; Abcam) overnight at 4 °C. After washing three times with PBS for 10 min, the appropriate Alexa Fluor™ 555-conjugated secondary antibody (Invitrogen, Carlsbad, CA, USA) was added and incubated for 1.5 h at RT. The nuclei of BV2 microglial cells were visualized using DAPI (Cat. No. D1306; Invitrogen).

### 4.12. Mouse Treatment

Wild-type mice (3-month-old C57BL/6 mice) were purchased from Hana Biotech (Pyeongtaek, Korea) and bred according to the Guide for the Care and Use of Laboratory Animals (8th edition) in a specific pathogen-free facility. Felodipine (Cat. No. F0814; Tokyo Chemical Industry, Tokyo, Japan) was prepared as a stock solution of 20 mg/mL in 50% ethanol and diluted to 1 mg/mL in vehicle (5% Tween-20 + 5% PEG300 in saline) before injection. Three different injection models were established. In the first model, to test the effects of felodipine on acute neuroinflammation in vivo, felodipine was intraperitoneally (i.p.) administered to C57BL/6 mice at a dose of 5 mg/kg/day for 3 days. LPS (Cat. No. L2630; Sigma-Aldrich, St. Louis, MO, USA) prepared in saline was intravenously (i.v.) injected 30 min after the last felodipine injection at a dose of 10 mg/kg. The second model was used to assess the effects of longer felodipine treatment on acute neuroinflammation; felodipine was injected daily for 7 days (i.p., 5 mg/kg/day), and on the 7th day, LPS (10 mg/kg, i.p.) was injected 30 min after the last felodipine injection. In the third model, to evaluate the effects of felodipine on chronic neuroinflammation-linked memory impairment, felodipine was injected daily for 9 days (5 mg/kg/day, i.p.), and LPS (250 μg/mL, i.p.) was injected 30 min after every felodipine injection. The C57BL/6 mice were sacrificed for analysis 8 h after the final injection. A total of 92 wild-type mice were used in our experiments, including replication experiments. No deaths were observed among the mice that received 10 mg/kg or 250 μg/mL LPS, and thus, no mice were excluded from the data analysis. All animal experiments were conducted under the supervision and approval of the Institutional Animal Care and Use Committee of KBRI (approval codes: IACUC-20-00024 and IACUC-21-00021).

### 4.13. Brain Tissue Preparation

Mouse brains were post-fixed in 4% PFA for 20 h at 4 °C, transferred to 30% sucrose solution for 3 days at 4 °C for cryoprotection and embedded in Tissue-Tek^®^ O.C.T. compound (Sakura Finetek, Torrance, CA, USA). The brains were then cryosectioned at a thickness of 30 μm using a Leica CM1860 cryostat (Leica Biosystems, Buffalo Grove, IL, USA) at −23 °C. The brain sections were stored in PBS with sodium azide (1%) at 4 °C until analysis.

### 4.14. Immunofluorescence Staining

Free-floating immunohistochemical staining was performed to immunolabel Iba-1, GFAP, COX-2, and IL-1β in brain tissues. Brain sections were briefly washed with PBS, and then, incubated with blocking solution (0.3% Triton X-100 + 0.05% bovine serum albumin in PBS) for 1 h at room temperature (RT). Subsequently, the brain sections were incubated overnight at 4°C with primary antibodies against Iba-1 (Cat. No. 019-19741; 1:500; Wako, Osaka, Japan), GFAP (Cat. No. AB5541; 1:500; Sigma-Aldrich), COX-2 (Cat. No. ab15191; Abcam, Cambridge, UK), and IL-1β (Cat. No. ab9722; Abcam) in blocking solution. The brain sections were washed three times with PBS for 10 min, and then, incubated for 2 h at RT with the appropriate Alexa Fluor™ 488- or 555-conjugated secondary antibody (Invitrogen, Carlsbad, CA, USA). The brain tissues were washed 4 times with PBS for 10 min and counterstained by adding 4′,6-diamidino-2-phenylindole (DAPI; Cat. No. D1306; Invitrogen) in the second washing step. Finally, the brain tissues were mounted on a glass slide and enclosed with VECTASHIELD*^®^* Antifade Mounting Medium (Vector Laboratories, Burlingame, CA, USA) before imaging using a DMi8 fluorescence microscope (Leica Microsystems, Wetzlar, Germany).

### 4.15. Y-Maze

The Y-maze test was performed on the 7th day of chronic LPS and felodipine administration to evaluate the effects of felodipine on short-term and spatial working memory impairments induced by LPS in C57BL/6 mice [97]. The mice were evaluated singly; one mouse explored the 3 arms (35 cm × 7 cm × 15 cm) of the maze, which met at a 120° angle, freely for 5 min. A SMART video camera recorded spontaneous alternations, which were then manually counted. The number of alternations was divided by the number of alternation triads to calculate the alternation percentage.

### 4.16. Novel Object Recognition (NOR) Test

The NOR test was performed on the 8th–9th days of chronic LPS and felodipine administration to determine if felodipine alters the impairments of long-term and recognition memory induced by LPS [97]. An open-field box (40 cm × 40 cm × 25 cm) was used as the apparatus for the NOR test. First, the mice were trained; a single mouse was placed in the box with two identical objects and permitted to explore the box for 5 min. To remove odor cues, 70% ethanol was used to thoroughly clean the box and objects between sessions. Twenty-four hours after training, the test was performed by returning the mouse to the box with one familiar object and one novel object. Counterbalancing of the locations of the familiar and novel objects was performed throughout the trials. The tests were recorded, and the exploration time was manually counted. Pointing of the mouse’s nose toward an object indicated exploratory behavior. Objects that were not touched for more than 7 s were excluded from statistics. Object preference (%) was calculated from the exploration times for the familiar and novel objects: [Object preference (%) = T Novel/(T Familiar + T Novel) × 100].

### 4.17. Golgi Staining

Golgi staining was performed after the behavior tests to determine the effects of felodipine on dendritic spine formation. For these experiments, we used the FD Rapid Golgi Stain kit (FD NeuroTechnologies, Columbia, MD, USA) as previously described [97]. Brains were dissected from the mice used in the behavior tests and immersed in solutions A and B for 2 weeks at room temperature. After subsequent immersion for 24 h in solution C at 4 °C, sectioning at a thickness of 150 μm was performed using a vibratome (VT1000S; Leica). Bright-field microscopy (Axioplan 2; Zeiss) images (at 63× magnification) were taken of cortical layer V and hippocampal CA1 pyramidal neurons and coded. The dendritic spine number was determined from the images in a blinded manner using ImageJ 7.4 (NIH, Bethesda, Maryland, U.S.). Only dendrites with a length greater than 20 μm were included in the dendritic spine number measurement.

### 4.18. Statistical Analysis

Data were analyzed using GraphPad Prism 7 software (GraphPad Software, San Diego, CA, USA). The unpaired two-tailed *t-test* with Welch’s correction was used for pairwise comparisons. One-way ANOVA and Tukey’s test were used for multiple comparisons. Significant differences were set at *p* < 0.05. Results are presented as means ± SDs (* *p* < 0.05, ** *p* < 0.01, *** *p* < 0.001, **** *p* < 0.0001). Statistical analysis are presented in Appendix A. 

## 5. Conclusions

The effects of L-type CCBs on neuroinflammation may differ depending on the specific drug and the microenvironment [78]. In this study, we demonstrated that the second-generation L-type CCB felodipine and first-generation L-type CCB nitrendipine inhibited LPS-induced proinflammatory responses by inhibiting TLR4/AKT/STAT3 signaling in BV2 microglial cells. In C57BL/6 mice, felodipine treatment suppressed LPS-stimulated microglial and astrocyte activation and proinflammatory cytokine COX-2 and IL-1β levels. More importantly, felodipine treatment reduced LPS-mediated deficits in spatial memory, dendritic spine formation, and microgliosis in a mouse model of chronic neuroinflammation induced by LPS. These results provide evidence supporting the repurposing of felodipine for neuroinflammation- and memory-associated diseases.

## Figures and Tables

**Figure 1 ijms-23-13606-f001:**
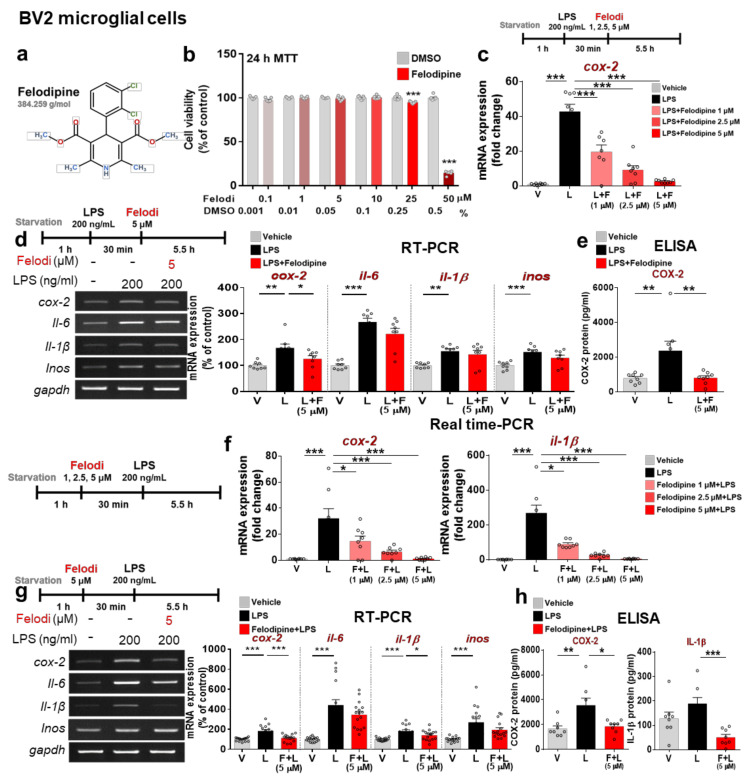
The L-type Ca^2+^ channel blocker felodipine suppresses LPS-induced proinflammatory cytokine levels in BV2 microglial cells. (**a**) Structure of felodipine. (**b**) MTT assay of the effect of felodipine on cell viability (felodipine: n = 7/group). (**c**) BV2 microglial cells were treated with LPS, followed by felodipine (1, 2.5, or 5 μM) as shown, and real-time PCR was performed (n = 8/group). (**d**) BV2 microglial cells were treated with LPS, followed by 5 μM felodipine as shown, and RT-PCR was conducted (n = 8/group). (**e**) ELISA analysis of the effect of felodipine post-treatment on secreted COX-2 levels (n = 8/group). (**f**) BV2 microglial cells were pre-treated with felodipine (1, 2.5, or 5 μM), followed by LPS as shown, and real-time PCR was performed (n = 8/group). (**g**) BV2 microglial cells were pre-treated with 5 μM felodipine, followed by LPS as shown, and RT-PCR was conducted (n = 16/group). (**h**) ELISA analysis of the effect of felodipine pre-treatment on secreted COX-2 and IL-1β levels (n = 8/group). * *p* < 0.05, ** *p* < 0.01, *** *p* < 0.001. V: vehicle; L: LPS; F or Felodi: felodipine.

**Figure 2 ijms-23-13606-f002:**
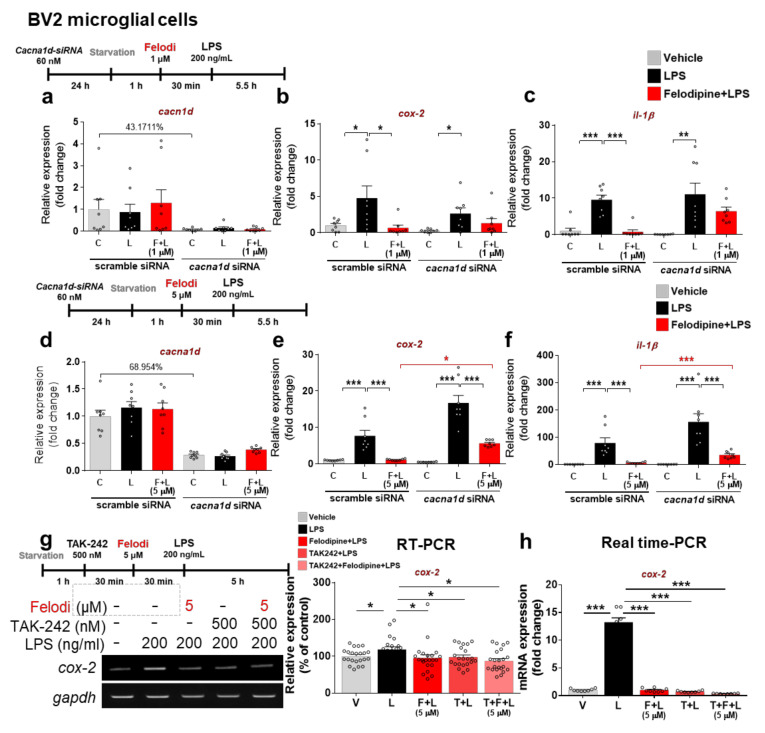
Felodipine downregulates LPS-stimulated proinflammatory responses through the L-type Ca^2+^ channel and TLR4 signaling in BV2 microglial cells. (**a**,**d**) Real-time PCR analysis of *cacna1d* gene expression in cells treated with 1 μM felodipine (**a**) or 5 μM felodipine (**d**), followed by LPS as shown (n = 8/group). (**b**,**c**,**e**,**f**) Real-time PCR analysis of *cox-2* and *il-1β* mRNA levels in cells transfected with *cacna1d* siRNA (60 nM) or scramble (control) siRNA for 24 h and subsequently treated with 1 μM felodipine (**b**,**c**) or 5 μM felodipine (**e**,**f**) and with LPS as shown (n = 8/group). (**g**,**h**) RT-PCR and real-time PCR analyses of proinflammatory cytokine *cox-2* expression in cells treated with TLR4 inhibitor (TAK-242), 5 μM felodipine, and LPS (felodipine: n = 22/group). * *p* < 0.05, ** *p* < 0.01, *** *p* < 0.001. V: vehicle; L: LPS; F or Felodi: felodipine.

**Figure 3 ijms-23-13606-f003:**
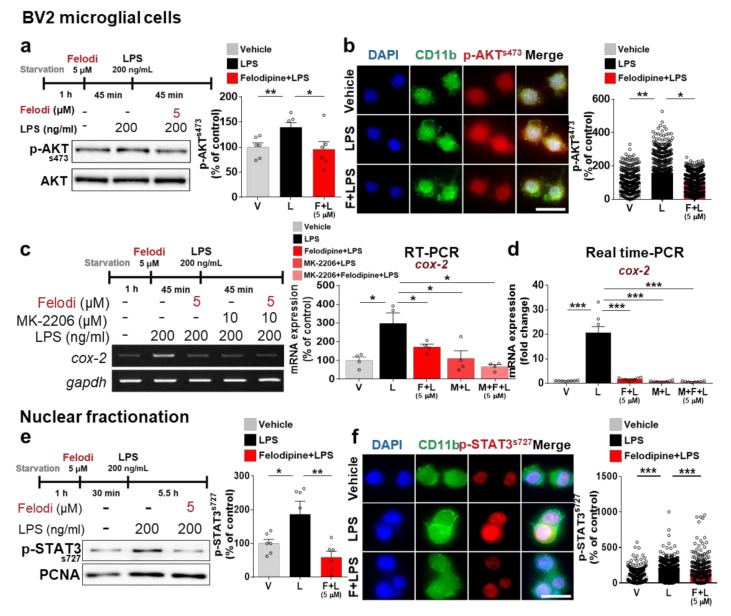
Felodipine inhibits LPS-stimulated AKT and STAT3 phosphorylation in BV2 microglial cells. (**a**) Western blot analysis of AKT phosphorylation in cells treated with 5 μM felodipine or vehicle (1% DMSO) for 45 min, followed by LPS (200 ng/mL) or PBS for 45 min (n = 6/group). (**b**) Immunocytochemical staining of p-AKT^S473^ in cells treated with 5 μM felodipine or vehicle (1% DMSO) for 45 min, followed by LPS (200 ng/mL) or PBS for 45 min (Veh, n = 443; LPS, n = 578; LPS+Felodi, n = 918). (**c**,**d**) Expression of the proinflammatory cytokine *cox-2* in cells treated as shown with an AKT inhibitor (MK-2206), 5 μM felodipine, and LPS and analyzed via RT-PCR (c, n = 4/group) or real-time PCR (d, n = 8/group). (**e**) Nuclear fractionation and Western blotting analysis of LPS-mediated nuclear p-STAT3^S727^ levels in cells treated with 5 μM felodipine or vehicle (1% DMSO) for 30 min, followed by LPS (200 ng/mL) or PBS for 5.5 h (n = 6/group). (**f**) Immunocytochemical staining analysis of LPS-induced nuclear STAT3 phosphorylation in cells treated with 5 μM felodipine or vehicle (1% DMSO) for 30 min, followed by LPS (200 ng/mL) or PBS for 5.5 h (Veh, n = 414; LPS, n = 326; Felodi+LPS, n = 308). * *p* < 0.05, ** *p* < 0.01, *** *p* < 0.001, scale bar: 20 μm. V: vehicle; L: LPS; F or Felodi: felodipine, M: MK-2206.

**Figure 4 ijms-23-13606-f004:**
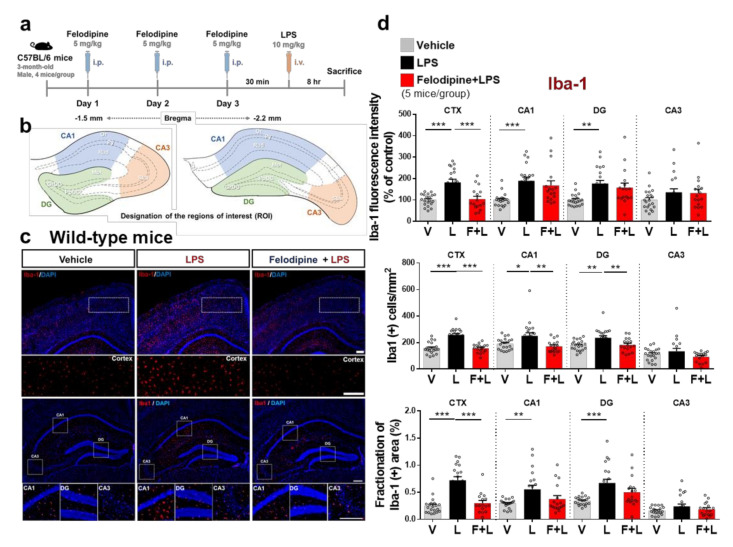
Daily administration of felodipine for 3 days decreases LPS-induced microgliosis in the cortex and hippocampus of C57BL/6 mice. (**a**) Experimental scheme of felodipine treatment in vivo. (**b**) Coordinates and specific locations of the hippocampus used for immunofluorescence staining. (**c**) C57BL/6 mice were injected with vehicle (5% Tween20 + 5% PEG300 in saline) or felodipine (5 mg/kg, i.p.) daily for 3 days. On day 3, LPS (10 mg/kg, i.v.) or PBS was injected, and immunofluorescence staining was conducted with an anti-Iba-1 antibody. (**d**) Quantification of data from (**c**) (n = 5 mice/group. Vehicle, n = 20 brain slices; LPS, n = 20 brain slices; felodipine+LPS, n = 16 brain slices. * *p* < 0.05, ** *p* < 0.01, *** *p* < 0.001, scale bar = 100 μM. V: vehicle; L: LPS; F or Felodi: felodipine.

**Figure 5 ijms-23-13606-f005:**
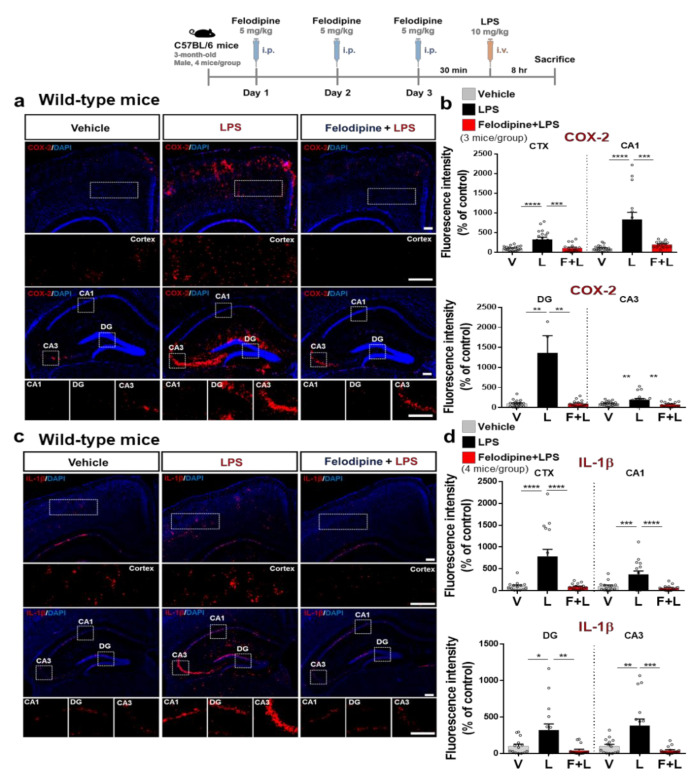
Daily administration of felodipine for 3 days diminishes LPS-mediated proinflammatory cytokine COX-2 and IL-1β levels in the brain in C57BL/6 mice. (**a**,**c**) Immunofluorescence staining of proinflammatory cytokine COX-2 and IL-1β expression in brain slices from C57BL/6 mice injected daily with vehicle (5% Tween-20 + 5% PEG300 in saline) or felodipine (5 mg/kg, i.p.) for 3 days, followed by LPS (10 mg/kg, i.v.) or PBS on day 3. (**b**,**d**) Quantification of data from (**a**,**c**) (n = 5 mice/group. Vehicle, n = 20 brain slices; LPS, n = 20 brain slices; felodipine+LPS, n = 16 brain slices. * *p* < 0.05, ** *p* < 0.01, *** *p* < 0.001, **** *p* < 0.0001, scale bar = 100 μM. V: vehicle; L: LPS; F or Felodi: felodipine.

**Figure 6 ijms-23-13606-f006:**
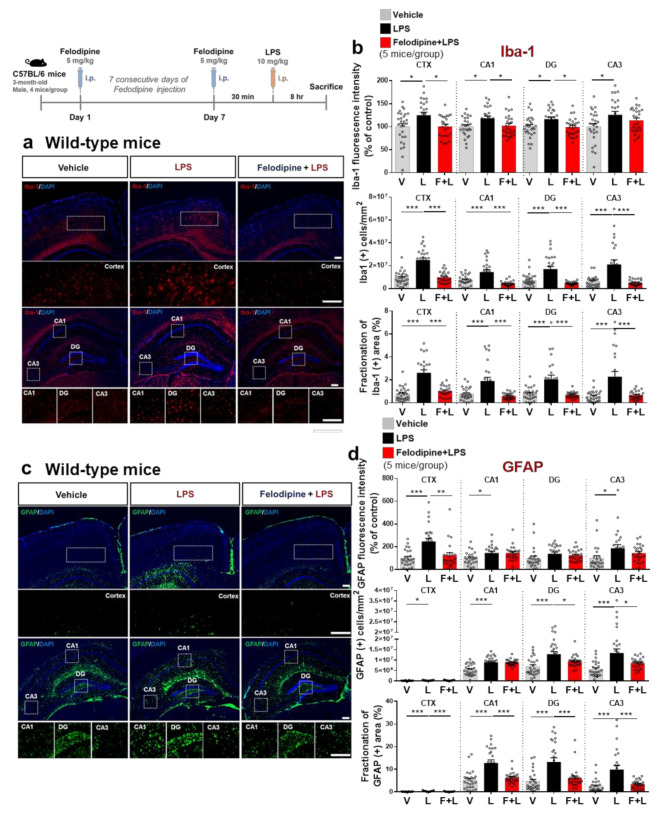
Daily administration of felodipine for 7 days suppresses LPS-stimulated microgliosis and astrogliosis in the brain in C57BL/6 mice. (**a**,**c**) Immunofluorescence staining of microglial and astroglial expression in brain slices from C57BL/6 mice injected daily with vehicle (5% Tween-20 + 5% PEG300 in saline) or felodipine (5 mg/kg, i.p.) for 7 days, followed by injection of LPS (10 mg/kg, i.p.) or PBS on day 7. (**b**,**d**) Quantification of data from (**a**,**c**) (n = 5 mice/group. Vehicle, n = 26 brain slices; LPS, n = 24 brain slices; felodipine+LPS, n = 26 brain slices). * *p* < 0.05, ** *p* < 0.01, *** *p* < 0.001, scale bar = 100 μM. V: vehicle; L: LPS; F or Felodi: felodipine.

**Figure 7 ijms-23-13606-f007:**
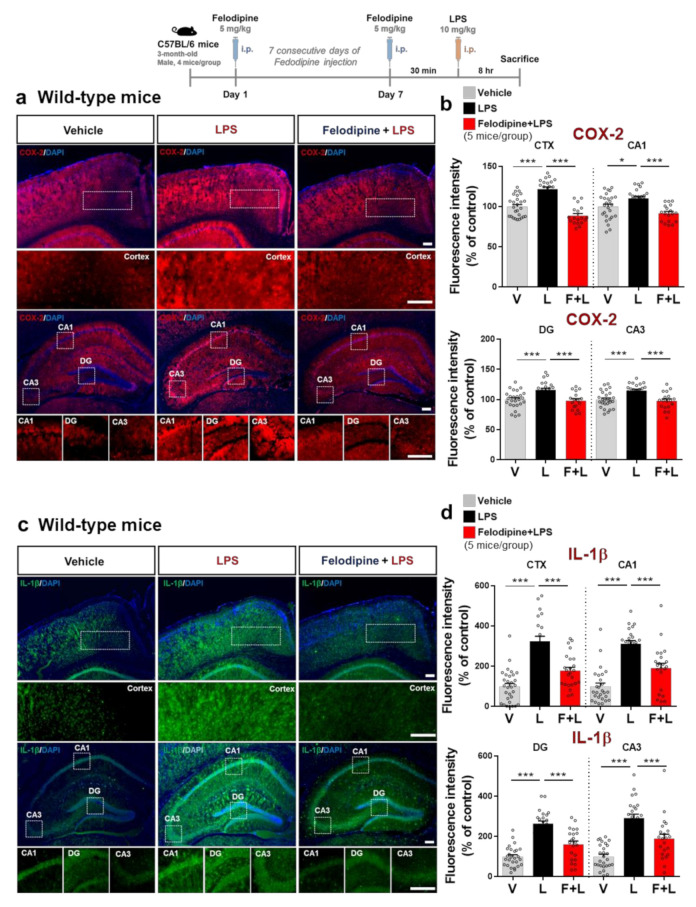
Daily administration of felodipine for 7 days downregulates LPS-induced proinflammatory cytokine COX-2 and IL-1β levels in C57BL/6 mice. (**a**,**c**) Immunofluorescence staining of proinflammatory cytokine COX-2 and IL-1β expression in brain slices from C57BL/6 mice injected daily with vehicle (5% Tween-20 + 5% PEG300 in saline) or felodipine (5 mg/kg, i.p.) for 7 days, followed by LPS (10 mg/kg, i.p.) or PBS on day 7. (**b**,**d**) Quantification of data from (**a**,**c**) (vehicle, n = 5 mice/group, n = 26 brain slices; LPS, n = 22 brain slices; felodipine+LPS, n = 18 brain slices. * *p* < 0.05, *** *p* < 0.001, scale bar = 100 μM. V: vehicle; L: LPS; F or Felodi: felodipine).

**Figure 8 ijms-23-13606-f008:**
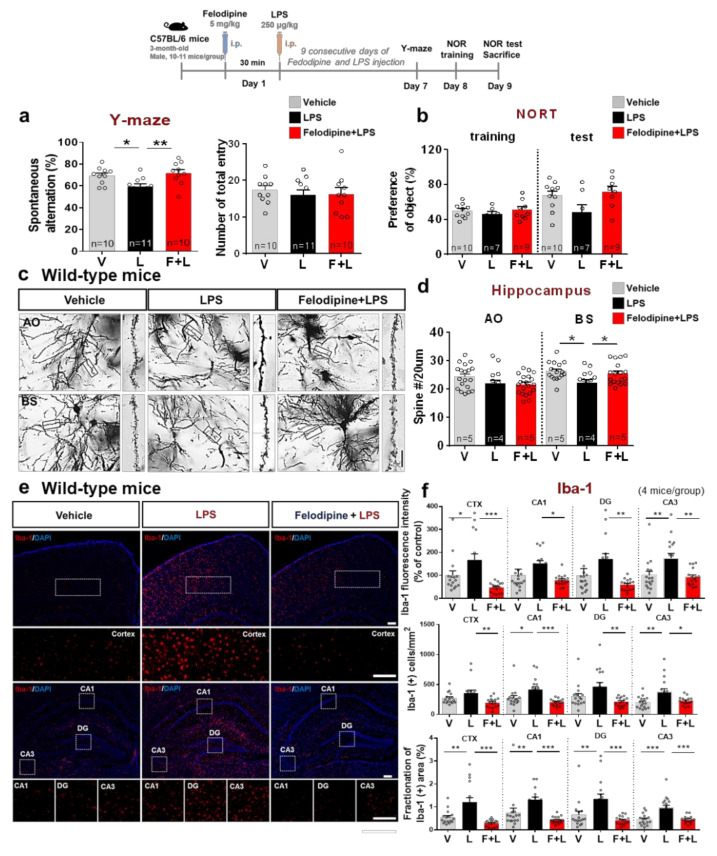
Treatment with felodipine daily for 9 days modulates LPS-induced short-term working memory impairment, hippocampal dendritic spine loss, and microgliosis in C57BL/6 mice. Three-month-old C57BL/6 mice were injected daily with vehicle (5% Tween-20 + 5% PEG300 in saline) or felodipine (5 mg/kg, i.p.), followed 30 min later by 250 μg/kg LPS or PBS for 9 days. (**a**) Y-maze tests were performed on day 7; spontaneous alternations and the number of total entries are shown (vehicle, n = 10 mice; LPS, n = 11 mice; felodipine+LPS, n = 10 mice). (**b**) Novel object recognition (NOR) training/tests were performed on days 8 and 9; object preference is shown (vehicle, n = 10 mice; LPS, n = 7 mice; felodipine+LPS, n = 9 mice). (**c**) Golgi staining was performed after the behavior experiments to measure dendritic spine number in the hippocampal AO and BS dendrites. (**d**) Quantification of data from (**c**) (vehicle, n = 19 slices from 5 mice; LPS, n = 15 slices from 4 mice; felodipine+LPS, n = 20 slices from 5 mice. (**e**) Microgliosis was assessed via immunofluorescence staining of Iba-1 in brain slices from the C57BL/6 mice used in the behavioral experiments. (**f**) Quantification of data from (**e**) (n = 4 mice/group, n = 16 brain slices). * *p* < 0.05, ** *p* < 0.01, *** *p* < 0.001. Golgi staining scale bar = 5 μM. IF scale bar = 100 μM. V: vehicle; L: LPS; F or Felodi: felodipine.

**Figure 9 ijms-23-13606-f009:**
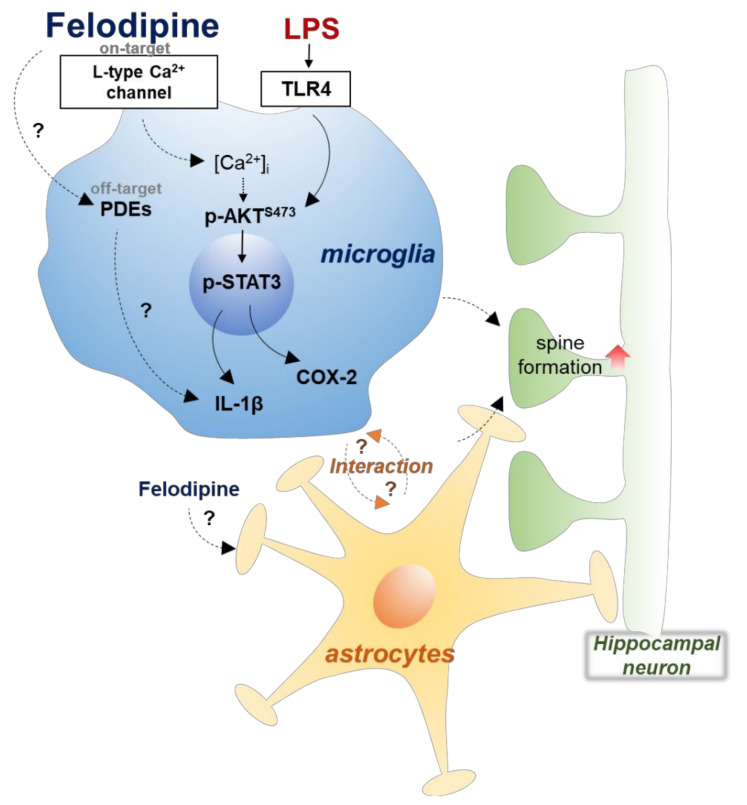
Schematic diagram of the effects of felodipine on LPS-induced neuroinflammatory responses and cognitive function. In microglial cells, felodipine downregulates LPS-induced proinflammatory cytokine COX-2 and IL-1β levels through L-type Ca^2+^ channel/TLR4/AKT signaling. Felodipine also inhibits LPS-stimulated nuclear STAT3 phosphorylation in microglia. In C57BL/6 (wild-type) mice, felodipine reduces LPS-induced micro/astrogliosis and proinflammatory cytokine COX-2 and IL-1β levels and reduces the short-term spatial memory impairment, decrease in hippocampal dendritic spinogenesis, and increase in microgliosis induced by chronic LPS administration. The capacity of felodipine to alter LPS-induced gliosis in vitro/in vivo and to modulate neuroinflammation-linked behavior suggests that felodipine may have potential for the treatment of neuroinflammation/cognitive function-linked diseases.

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
