# Peer review of "L-Type Ca2+ Channel Inhibition Rescues the LPS-Induced Neuroinflammatory Response and Impairments in Spatial Memory and Dendritic Spine Formation"

_ijms, 2022, doi:10.3390/ijms232113606_

Round 1

Reviewer 1 Report

In this study, Kim et al. investigated the effects of L-type Ca++ channel blockers, namely felodipine and nitrendipine on LPS-induced neuroinflammation, by in vitro and in vivo approaches. Data is of interest and bring interesting evidence about the effects of both drugs, mainly felodipine. However, there are some points that need clarification to allow the paper publication.

1.     The authors must provide additional information regarding the bacterial source and the serotype of LPS that has been used in the study. This is very relevant if one would like to reproduce the experimental model.

2.     The authors are required to provide the final concentrations of DMSO used for in vitro experiments in the item 2.2. I believe that 1% is the correct, but this has been informed only latter in the manuscript.

3.     There is no rational for the concentration range that had been used for the calcium channel blockers. The authors have tested different concentrations in the MTT assay, but this is not enough to justify the choose for 5 mM for further functional studies. At least one experiment testing three concentrations or even previous literature data are required for justifying the selected concentration as effective. It is possible that the absence of effects in some inflammatory parameters are related with the use of a low concentration, instead of a specific mechanism of the drugs.

4.     Still concerning the first item, it is not clear whether the in vivo doses of felodipine have been chosen. Please, provide references for such dose, as well as for the route of administration. What about the effects of felodipine given by oral route? The justification provided in the discussion based on clinical doses is not appropriate, except if the authors have made an extrapolation from human to mouse doses by considering the surface area.

5.     The in vitro experiments using pre- and post-treatment schemes are not totally clear and should be further explored in the discussion. In clinical set, neuroinflammation is supposed to be installed and cannot be prevented, so the calcium blockers are expected to be effective if given therapeutically, rather than in a prophylactic schedule.

6.     This reviewer is confused with the abbreviation “i.v.” for LPS administration – is this intravenous or intraventricular? Both are possible in view of the study aims; please, clarify it.

7.     For ex vivo experiments, the number of animals in the group rather than the number of replicates or slices should be included in the graphs, as well as in the statistical analysis.

8.     Information regarding the total number of animals and exclusion of mice during the experiments should be provided. A dose of 10 mg/kg of LPS can lead to death – this comment is also pertinent regarding the chronic administration of LPS at 250 mg/kg. The number of animals that had been lost should be provided as a transparency criterion. Moreover, the sample calculation size should be provided for in vivo experiments.

9.     There is no information in the statistical analysis regarding the test that had been adopted to determine data Gaussian distribution. If behavioral data does not show normality distribution, different statistical tests should be carried out.

10.  Data on the relevance of TLR-4 in felodipine effects sounds highly fragile. Without WB experiments showing receptors expression, plus additional protocols for showing receptor activation, the author’s conclusion are quite speculative.  

11.  The absence of effects for NF-kB activation is curious and deserve further discussion.

12.  If possible, please include further data from the in vivo experiments, such as changes in body temperature and general locomotion activity. Is it possible to consider that part of the effects of felodipine rely on peripheral rather than central mechanisms? Please, discuss this point in the revised version of the manuscript.

Author Response

Responses to Reviewer 1 (please see the attached file)

We would like to thank the reviewer for their thoughtful suggestions and helpful comments. We have revised the manuscript to address the reviewer’s concerns, and our point-by-point responses to each of the reviewer’s comments are provided below. In the revised manuscript, changes are shown in red font. 

Reviewer 1

In this study, Kim et al. investigated the effects of L-type Ca++ channel blockers, namely felodipine and nitrendipine on LPS-induced neuroinflammation, by in vitro and in vivo approaches. Data is of interest and bring interesting evidence about the effects of both drugs, mainly felodipine. However, there are some points that need clarification to allow the paper publication.

  1. The authors must provide additional information regarding the bacterial source and the serotype of LPS that has been used in the study. This is very relevant if one would like to reproduce the experimental model.

Response: We apologize for any confusion. We added the information for LPS in section 2.5 of the Materials and methods section (please see line 117).

  1. The authors are required to provide the final concentrations of DMSO used for in vitro experiments in the item 2.2. I believe that 1% is the correct, but this has been informed only latter in the manuscript.

Response: We apologize for any confusion. We added the information of final concentration of DMSO used in vitro experiment throughout the manuscript (Please see line 90).

  1. There is no rational for the concentration range that had been used for the calcium channel blockers. The authors have tested different concentrations in the MTT assay, but this is not enough to justify the choose for 5 mM for further functional studies. At least one experiment testing three concentrations or even previous literature data are required for justifying the selected concentration as effective. It is possible that the absence of effects in some inflammatory parameters are related with the use of a low concentration, instead of a specific mechanism of the drugs.

Response: We thanks to reviewer’s insightful comments. In accordance with the reviewer’s suggestion, we conducted additional experiments using three concentrations of felodipine (1, 2.5 or 5 mM) to assess the effects of felodipine on LPS-induced proinflammatory cytokine levels in BV2 microglial cells by real-time PCR. To examine the effects of post-treatment with felodipine on LPS-mediated proinflammatory cytokine levels, BV2 microglial cells were treated with 200 ng/mL LPS or PBS for 30 min followed by felodipine (1, 2.5, or 5 μM) or vehicle (1% DMSO) for 5.5 h. Subsequent real-time PCR showed that post-treatment with felodipine significantly reduced LPS-evoked cox-2 mRNA levels in BV2 microglial cells in a dose-dependent manner (Fig. 1c).

Next, we investigated the effects of pre-treatment with felodipine on LPS-evoked proinflammatory cytokine levels in vitro. BV2 microglial cells were treated with felodipine (1, 2.5, or 5 μM) or vehicle (1% DMSO) for 30 min followed by 200 ng/mL LPS or PBS for 5.5 h. Real time-PCR showed that pre-treatment with felodipine significantly reduced LPS-stimulated cox-2 and il-1β mRNA levels in a dose-dependent manner (Fig. 1f). Based on these findings, we selected a felodipine concentration of 5 μM for further experiments/analyses. We added these new findings in Figure 1c and 1f and revised the Results section in the revised manuscript (please see lines 323-329, 338-342, and Figure 1).

  1. Still concerning the first item, it is not clear whether the in vivo doses of felodipine have been chosen. Please, provide references for such dose, as well as for the route of administration. What about the effects of felodipine given by oral route? The justification provided in the discussion based on clinical doses is not appropriate, except if the authors have made an extrapolation from human to mouse doses by considering the surface area.

Response: We thank the reviewer for these insightful comments. To test the effects of felodipine on LPS-induced neuroinflammation in vivo, we chose intraperitoneal (i.p.) injection of 5 mg/kg of felodipine. According to the literature, 5 mg/kg/day of felodipine (i.p.) effectively reduces neurodegenerative disease-associated abnormal aggregation in the brain [1]. Specifically, 5 mg/kg/day felodipine (i.p.) significantly reduces α-synuclein aggregated neurons in the piriform cortex and motor cortex in mRFP-GFP-LC3-labelled mice [1]. In the same study, the authors investigated the pharmacokinetic (PK) parameters and mouse brain/plasma concentrations after single i.p. injection of 5 mg/kg felodipine and found that 5 mg/kg felodipine effectively crosses the blood-brain barrier (please see Tables 2 and 3 in the reference as we attached). Therefore, we chose a felodipine dose of 5 mg/kg (i.p. injection). This information has been added in the revised manuscript (please see lines 484-486). 

(Siddiqi et al., 2019, Nature Communications)

Felodipine has been reported to have low oral bioavailability (<15%) due to its poor water solubility and high first pass metabolism [2]. In a surface science (dynamic surface tension) study, Koli and co-workers investigated the ability of microemulsions to enhance the permeability and oral bioavailability of felodipine. They found that a microemulsion comprising Capmul MCM, Tween 20 and polyethylene glycol significantly increased the permeability of felodipine (74.1% after 1h) compared with a felodipine suspension (16.1% after 1 h) in ex vivo; however, the bioavailability of the felodipine microemulsion in rats was low (relative bioavailability = 21.9) [2]. Given this low oral bioavailability, we selected the intraperitoneal route for the in vivo studies of felodipine in this study. 

  1. The in vitro experiments using pre- and post-treatment schemes are not totally clear and should be further explored in the discussion. In clinical set, neuroinflammation is supposed to be installed and cannot be prevented, so the calcium blockers are expected to be effective if given therapeutically, rather than in a prophylactic schedule.

Response: We thank the reviewer for these insightful comments. Our first objective in this study was to determine whether felodipine pre-treatment (to investigate the preventive effects of felodipine on neuroinflammatory responses) or post-treatment (to examine the curative effects of felodipine on neuroinflammatory responses) could regulate LPS-induced proinflammatory responses in vitro. Interestingly, we observed that felodipine pre-treatment downregulated LPS-induced proinflammatory responses more effectively than felodipine post-treatment in vitro. Therefore, we further investigated the effects of felodipine pre-treatment on LPS-stimulated neuroinflammation in vivo.

A previous study reported that post-treatment with the L-type CCB Zinc20267861 significantly reduces microglial activation in a mouse model of suture-induced inflammation [3]. Another study demonstrated that post-treatment with the L-type CCB nimodipine significantly suppresses LPS-induced proinflammatory cytokine levels and Iba-1-positive cells in the hippocampus in mice [4]. These observations and our findings suggest that both pre- and post-treatment with L-type CCBs modulate neuroinflammatory responses in vivo. We will further address the effects of felodipine post-treatment on LPS-mediated neuroinflammatory responses in vivo in future work.

We added this in the Discussion sections in the revised manuscript (please see line 813-825).

  1. This reviewer is confused with the abbreviation “i.v.” for LPS administration – is this intravenous or intraventricular? Both are possible in view of the study aims; please, clarify it.

Response: We apologize for any confusion. The abbreviation “i.v.” means intravenous injection; we added this definition in the manuscript (please see line 224 and 491).

  1. For ex vivo experiments, the number of animals in the group rather than the number of replicates or slices should be included in the graphs, as well as in the statistical analysis.

Response: We appreciate the reviewer’s suggestion. To address this issue, we added the number of animals in each figure and included a table of the statistical analyses in the Supplementary Table 1 and 2.

  1. Information regarding the total number of animals and exclusion of mice during the experiments should be provided. A dose of 10 mg/kg of LPS can lead to death – this comment is also pertinent regarding the chronic administration of LPS at 250 mg/kg. The number of animals that had been lost should be provided as a transparency criterion. Moreover, the sample calculation size should be provided for in vivo experiments.

Response: We appreciate the reviewer’s suggestion. As requested, we added the total number of animals used in each experiment in the Materials and methods section and Figure legends. In addition, we added the following sentence in the Materials and methods section: “No deaths were observed among the mice that received 10 mg/kg or 250 μg/ml LPS, and thus no mice were excluded from the data analysis.” (Please see lines 232-235). Finally, the sample sizes are provided in the figure legends, and tables of the statistical analyses are provided as Supplementary tables 1 and 2 in the revised Supplementary materials.

  1. There is no information in the statistical analysis regarding the test that had been adopted to determine data Gaussian distribution. If behavioral data does not show normality distribution, different statistical tests should be carried out.

Response: We appreciate the reviewer’s suggestion. To address this comment, we added statistical information in all figures (including behavioral data with a normal distribution) and in the revised Supplementary materials (please see Supplementary table 2, the Y-maze tests in Figure 8a, and the NOR tests in Figure 8b).

  1. Data on the relevance of TLR-4 in felodipine effects sounds highly fragile. Without WB experiments showing receptors expression, plus additional protocols for showing receptor activation, the author’s conclusion are quite speculative.  

Response: We thank the reviewer for these insightful comments. To address the reviewer’s concern, we conducted additional experiments using cell surface biotinylation to measure the effects of felodipine on cell surface levels of TLR4. For these experiments, BV2 microglial cells were treated with 5 μM felodipine or vehicle (1% DMSO) for 30 min followed by LPS (200 ng/ml) or PBS for 5.5 h, and surface proteins were labeled with EZ-Link Sulfo-NHS-SS-Biotin. The biotin-labelled surface proteins were isolated and immobilized by incubation with NeutroAvidin™ gel beads for 1 h at room temperature. The biotinylated proteins were then eluted, and surface proteins were analyzed by western blotting with an antibody recognizing the N-terminus of TLR4. We found that LPS treatment increased the cell-surface levels of TLR4 in BV2 microglial cells compared with the control treatment (Please see the figure as blows). In addition, a trend toward decreased LPS-induced cell surface levels of TLR4 was observed in BV2 microglial cells treated with felodipine and LPS compared with cells treated with LPS, but the total TLR4 level was not altered (Please see the figure as below). Based on our findings, we now toned down the relevance of TLR4 in felodipine in the revised manuscript.

  1. The absence of effects for NF-kB activation is curious and deserve further discussion.

Response: We thank the reviewer for this advice. We added a further discussion of the absence of effects of felodipine on NF-κB in the Discussion section in the revised manuscript (please see lines 758-769).

  1. If possible, please include further data from the in vivo experiments, such as changes in body temperature and general locomotion activity. Is it possible to consider that part of the effects of felodipine rely on peripheral rather than central mechanisms? Please, discuss this point in the revised version of the manuscript.

Response: We appreciate the reviewer’s suggestion. Unfortunately, we do not have data for body temperature and general locomotor activity. However, as the reviewer suggested, we discussed the possibility that the effects of felodipine rely on peripheral rather than central mechanisms in the Discussion section in the revised manuscript (please see lines 787-797).

We believe that we have fully addressed the reviewer’s concerns and that the manuscript is stronger as a result. We hope that the revised manuscript is now suitable for publication in IJMS.

References

  1. Siddiqi, F. H.; Menzies, F. M.; Lopez, A.; Stamatakou, E.; Karabiyik, C.; Ureshino, R.; Ricketts, T.; Jimenez-Sanchez, M.; Esteban, M. A.; Lai, L.; Tortorella, M. D.; Luo, Z.; Liu, H.; Metzakopian, E.; Fernandes, H. J. R.; Bassett, A.; Karran, E.; Miller, B. L.; Fleming, A.; Rubinsztein, D. C., Felodipine induces autophagy in mouse brains with pharmacokinetics amenable to repurposing. Nat Commun 2019, 10, (1), 1817.
  2. Koli, A. R.; Ranch, K. M.; Patel, H. P.; Parikh, R. K.; Shah, D. O.; Maulvi, F. A., Oral bioavailability improvement of felodipine using tailored microemulsion: Surface science, ex vivo and in vivo studies. Int J Pharm 2021, 596, 120202.
  3. Saddala, M. S.; Lennikov, A.; Mukwaya, A.; Yang, Y.; Hill, M. A.; Lagali, N.; Huang, H., Discovery of novel L-type voltage-gated calcium channel blockers and application for the prevention of inflammation and angiogenesis. J Neuroinflammation 2020, 17, (1), 132.
  4. Hopp, S. C.; D'Angelo, H. M.; Royer, S. E.; Kaercher, R. M.; Crockett, A. M.; Adzovic, L.; Wenk, G. L., Calcium dysregulation via L-type voltage-dependent calcium channels and ryanodine receptors underlies memory deficits and synaptic dysfunction during chronic neuroinflammation. J Neuroinflammation 2015, 12, 56.

Reviewer 2 Report

In the paper “Inhibition of L-type Ca2+ channel rescues the LPS-induced neuroinflammatory response and impairments in spatial memory and dendritic spine formation”, authors describe the use of CCB such as felodipine and nitrendipine to inhibit LPS-evoked pro-inflammatory cytokine expression.

The article is interesting; however, there are several points to clarify.

Major comments

Authors assert that the inhibitory effect of felodipine and nitrendipine occurs by the inhibition of L-type channels. This affirmation is not true because in LPS treated cells the silencing of L-type channel main sub-unit does not reduces the expression of both cox-2 and il-1b compared with control cells treated with LPS, indeed it seems to increase.

No channel activity analysis after cacna 1 d silencing or CCB treatment were performed. How can say the Authors that the calcium signal is impaired by CCB treatment or channel silencing if they do not measure the activity of these channels?

In my opinion, data shown in this paper indicate that CCB treatment act in a L-type channel independent manner. Therefore, or Author demonstrate the involvement of L-type channel in the signal linked to LPS or they have to change their conclusions.

The expression of cox-2 and il-1b shown in figures 2 bc and 2 ef in cells treated with scramble sequences and LPS is different. How do the Authors explain this discrepancy?

In figure 2g the treatment with LPS does not increase cox-2 expression as shown previously. This figure may be improved.

The forth bar in both figures 2 g and 2 h is incorrect. T+F and T+N has to be substituted with T+L. Same thing for figure 3d. M+F becomes M+L.

In Real Time PCR era, quantifications have to be performed by this technique.

Minor criticisms

The involvement of TLR4 in the inflammatory response is demonstrated by the reduction of cox-2 and il-1b expression in LPS treated cells after application of the specific inhibitor. This should be emphasized in the text. Same thing with the inhibition of Akt by MK-2206.

The Discussion should be shorter. It is too much long.

I recommend a little revision of English

Author Response

Responses to Reviewer 2 (Please see the attached file!)

We would like to thank the reviewer for their thoughtful suggestions and helpful comments. We have revised the manuscript to address the reviewer’s concerns, and our point-by-point responses to each of the reviewer’s comments are provided below. In the revised manuscript, changes are shown in red font.

Reviewer 2

In the paper “Inhibition of L-type Ca2+ channel rescues the LPS-induced neuroinflammatory response and impairments in spatial memory and dendritic spine formation”, authors describe the use of CCB such as felodipine and nitrendipine to inhibit LPS-evoked pro-inflammatory cytokine expression.

The article is interesting; however, there are several points to clarify.

Major comments

Authors assert that the inhibitory effect of felodipine and nitrendipine occurs by the inhibition of L-type channels. This affirmation is not true because in LPS treated cells the silencing of L-type channel main subunit does not reduces the expression of both cox-2 and il-1b compared with control cells treated with LPS, indeed it seems to increase.

Response: We thank the reviewer for these insightful comments. In the initial submission, we reported that 5 μM felodipine significantly reduced the LPS-induced increases in cox-2 and il-1β mRNA levels compared with LPS treatment (Fig. 2e-f). However, in cells in which cacna1d (the main subunit of the L-type Ca2+ channel) was knocked down, 5 μM felodipine and LPS significantly increased cox-2 and il-1β mRNA levels compared with cells transfected with scramble-siRNA and treated with 5 μM felodipine and LPS (Fig. 2e-f). Similar results were obtained when 5 μM felodipine was replaced with 5 μM nitrendipine (Supplementary Fig. 2a-c). Based on our findings, we suggested that the suppression of LPS-induced cox-2 and il-1β mRNA levels by 5 μM felodipine was partially dependent on the L-type Ca2+ channel.

During the revision, we conducted additional experiments to examine whether a low dose of felodipine would differentially regulate LPS-induced proinflammatory responses in cells in which cacna1d was knocked down by siRNA. For this experiment, BV2 microglial cells were transfected with cacna1d siRNA (60 nM) or scramble (control) siRNA for 24 h and then treated sequentially with 1 μM felodipine or 1% DMSO and with LPS (200 ng/ml) or PBS. Subsequent real-time PCR showed that in BV2 microglial cells in which cacna1d was knocked down, 1 μM felodipine did not significantly alter LPS-induced cox-2 and il-1β mRNA levels (Fig. 2b-c). These data indicate that the suppression of LPS-induced cox-2 and il-1β mRNA levels by 1 μM felodipine is dependent on the L-type Ca2+ channel, whereas the effects of 5 μM felodipine are only partially dependent on the L-type Ca2+ channel.

Why do high and low doses of felodipine differentially regulate LPS-induced proinflammatory responses? It is possible that 5 μM felodipine completely blocks the L-type Ca2+ channel, leading to further inhibition of off-targets of felodipine (e.g., PDE). A few studies have reported that felodipine binds and inhibits calcium-binding proteins such as calmodulin, calmodulin-dependent cyclic nucleotide phosphodiesterase (PDE1), and the voltage-gated T-type calcium channel [1, 2]. Interestingly, PDE inhibitors can suppress microglial activation and proinflammatory cytokines in vivo and in vitro [3, 4]. In addition, it is reported that antisense oligos (ASOs) significantly reduce its on-target (BACH1) and off-target (SRRM2, TLE3, USP9X) mRNAs in a dose- and time-dependent manner in vitro, indicating that drug can inhibit it’s off-targets in a dose-dependently [5]. In the context of these observations in the literature, our findings suggest that felodipine differentially suppresses LPS-induced neuroinflammation by inhibiting the L-type Ca2+ channel (on-target) and/or other proteins (i.e., off-target, such as PDE) in a dose-dependent manner in vitro and in vivo.

We added our new data in the revised Figure 2a-c and the Results and Discussion sections of the revised manuscript (please see lines 367-387).

No channel activity analysis after cacna 1 d silencing or CCB treatment were performed. How can say the Authors that the calcium signal is impaired by CCB treatment or channel silencing if they do not measure the activity of these channels?

Response: We thank the reviewer for these insightful comments. It is well known that felodipine blocks the L-type Ca2+ channel directly in vivo and in vitro [6-9]. For instance, a recent study found that knockdown of the Cav1.2 subunit of the L-type Ca2+ channel decreases Ca2+ influx by ~80% in vitro [10]. In cultured microglia, the L-type CCB nifedipine significantly blocks intracellular Ca2+ influx [11]. Based on these observations, we expect that the L-type CCB felodipine inhibits Ca2+ influx to alter LPS-mediated proinflammatory responses via the L-type Ca2+ channel. In fact, our additional results showing that proinflammatory cytokines are reduced in BV2 microglial cells after treatment with a low dose of felodipine (1 μM) provide indirect evidence that the L-type Ca2+ channel is functionally inhibited. Unfortunately, we were unable to perform L-type Ca2+ channel activity experiments due to the limitations of our laboratory equipment and the deadline for resubmission.

In my opinion, data shown in this paper indicate that CCB treatment act in a L-type channel independent manner. Therefore, or Author demonstrate the involvement of L-type channel in the signal linked to LPS or they have to change their conclusions.

Response: We thank the reviewer for these insightful comments. In our additional experiments (Figure 2a-c), we found that 1 and 5 μM felodipine differentially regulate LPS-mediated proinflammatory responses through the L-type CCB and/or off-targets in vitro.

In the initial submission, we reported that 5 μM felodipine significantly reduced the LPS-induced increases in cox-2 and il-1β mRNA levels compared with LPS treatment (Fig. 2e-f). However, in cells in which cacna1d (the main subunit of the L-type Ca2+ channel) was knocked down, 5 μM felodipine and LPS significantly increased cox-2 and il-1β mRNA levels compared with cells transfected with scramble-siRNA and treated with 5 μM felodipine and LPS (Fig. 2e-f). These data suggest that the effects of 5 μM felodipine on LPS-evoked proinflammatory responses are partially dependent on the L-type Ca2+ channel. We then discuss several possible molecular mechanisms by which 5 μM felodipine regulates LPS-induced proinflammatory cytokine release. One possibility is that 5 μM felodipine completely blocks the L-type Ca2+ channel, leading to further inhibition of off-targets of felodipine (e.g., PDE). As a result, effects of felodipine on LPS-mediated proinflammatory responses were observed in the absence of cacan1d in BV2 microglial cells.

To further confirm our findings, we conducted additional experiments to assess the effects of 1 μM felodipine on LPS-stimulated proinflammatory responses through the L-type Ca2+ channel. Importantly, 1 μM felodipine did not significantly alter LPS-induced cox-2 and il-1β mRNA levels, indicating that 1 μM felodipine downregulates LPS-stimulated proinflammatory cytokine levels in an L-type Ca2+ channel-dependent manner (Figure 2b-c). In addition, it is reported that antisense oligos (ASOs) significantly reduce its on-target (BACH1) and off-target (SRRM2, TLE3, USP9X) mRNAs in a dose- and time-dependent manner in vitro, indicating that drug can inhibit it’s off-targets in a dose-dependently [5]. In the context of these observations in the literature, our findings suggest that felodipine differentially suppresses LPS-induced neuroinflammation by inhibiting the L-type Ca2+ channel (on-target) and/or other proteins (i.e., off-target, such as PDE) in a dose-dependent manner in vitro and in vivo.

We revised our manuscript and added the new data. Moreover, we offer potential explanations of the results in the Discussion section in the revised manuscript (please see lines 367-387, 704-725 in the Results and Discussion sections of the revised manuscript).

The expression of cox-2 and il-1b shown in figures 2 bc and 2 ef in cells treated with scramble sequences and LPS is different. How do the Authors explain this discrepancy?

Response: We thank the reviewer for this insightful comment. To address the reviewer’s concern, we conducted a statistical analysis of the data for scramble-siRNA and cacan1d-siRNA cells treated with LPS in Figure 2e, 2f and supplementary 2b, 2c. One-way ANOVA indicated that there was no significant difference between the scramble siRNA-LPS treatment and the cacn1d siRNA-LPS treatment. As the difference is not significant, our main findings in Figure 2 are unaffected. To further support our findings, please see the detailed information in the following table:

Figure 2e

Tukey's multiple comparisons test

Mean Diff.

95.00% CI of diff.

Significant?

Summary

Adjusted P Value

L vs. L

-9.014

-19.09 to 1.064

No

ns

0.0890

Figure 2f

Tukey's multiple comparisons test

Mean Diff.

95.00% CI of diff.

Significant?

Summary

Adjusted P Value

L vs. L

-77.15

-221 to 66.7

No

ns

0.4099

Supplementary Figure 2b

Tukey's multiple comparisons test

Mean Diff.

95.00% CI of diff.

Significant?

Summary

Adjusted P Value

L vs. L

5.137

-1.542 to 11.82

No

ns

0.2174

Supplementary Figure 2c

Tukey's multiple comparisons test

Mean Diff.

95.00% CI of diff.

Significant?

Summary

Adjusted P Value

L vs. L

44.47

-10.99 to 99.92

No

ns

0.1818

In figure 2g the treatment with LPS does not increase cox-2 expression as shown previously. This figure may be improved.

Response: We thank the reviewer for this comment. To address the reviewer’s concern, we conducted additional real-time PCR experiments to further confirm our RT-PCR results in Figure 2g. BV2 microglial cells were treated successively with (i) 500 nM TLR4 inhibitor (TAK-242) or 1% DMSO (vehicle) for 30 min; (ii) 5 μM felodipine or vehicle for 30 min; and (iii) 200 ng/mL LPS or PBS for 5 h. Subsequent real-time PCR showed that treatment with TAK-242, felodipine, and LPS did not significantly affect LPS-evoked cox-2 mRNA levels compared with treatment with felodipine and LPS or treatment with TAK-242 and LPS (Figure 2i). These results suggest that the inhibitory effect of felodipine on LPS-induced cox-2 expression is dependent on TLR4 signaling. We added these results in Figure 2i and revised the text in the manuscript (please see line 398-401).

The forth bar in both figures 2 g and 2 h is incorrect. T+F and T+N has to be substituted with T+L. Same thing for figure 3d. M+F becomes M+L.

Response: We apologize for any confusion. We revised the labeling in Figures 2g, 2h and 3d.

In Real Time PCR era, quantifications have to be performed by this technique.

Response: To address the reviewer’s comment, we conducted additional q-PCR experiments as shown in Figures 1c, 1f, 2a-c, 2h, and 3d. In addition, new data were incorporated into the figure legends, Figures 1c, 1f, 2a-c, 2i, and 3d, and the Results section of the revised manuscript (please see lines 323-327, 337-342, 366-387, 398-401, and 439-442).

Minor criticisms

The involvement of TLR4 in the inflammatory response is demonstrated by the reduction of cox-2 and il-1b expression in LPS treated cells after application of the specific inhibitor. This should be emphasized in the text. Same thing with the inhibition of Akt by MK-2206.

Response: We thank the reviewer for these insightful comments. The involvement of TLR4 and AKT signaling after treatment with specific inhibitors is discussed and emphasized in the revised manuscript (please see lines 726-736, and 738-750).

The Discussion should be shorter. It is too much long.

Response: We thank the reviewer for this advice, and we have revised the Discussion section accordingly.

I recommend a little revision of English

Response: We appreciate the reviewer’s comment. To address the reviewer’s suggestion, the entire manuscript was revised, double-checked by the authors and edited by qualified experts.

We believe that we have fully addressed the reviewer’s concerns and that the manuscript is stronger as a result. We hope that the revised manuscript is now suitable for publication in IJMS.

References

  1. Sharma, R. K.; Wang, J. H.; Wu, Z., Mechanisms of inhibition of calmodulin-stimulated cyclic nucleotide phosphodiesterase by dihydropyridine calcium antagonists. J Neurochem 1997, 69, (2), 845-50.
  2. Walsh, M. P.; Sutherland, C.; Scott-Woo, G. C., Effects of felodipine (a dihydropyridine calcium channel blocker) and analogues on calmodulin-dependent enzymes. Biochem Pharmacol 1988, 37, (8), 1569-80.
  3. Kim, D. Y.; Park, J. S.; Leem, Y. H.; Park, J. E.; Kim, H. S., The Potent PDE10A Inhibitor MP-10 (PF-2545920) Suppresses Microglial Activation in LPS-Induced Neuroinflammation and MPTP-Induced Parkinson's Disease Mouse Models. J Neuroimmune Pharmacol 2021, 16, (2), 470-482.
  4. Zang, J.; Wu, Y.; Su, X.; Zhang, T.; Tang, X.; Ma, D.; Li, Y.; Liu, Y.; Weng, Z.; Liu, X.; Tsang, C. K.; Xu, A.; Lu, D., Inhibition of PDE1-B by Vinpocetine Regulates Microglial Exosomes and Polarization Through Enhancing Autophagic Flux for Neuroprotection Against Ischemic Stroke. Front Cell Dev Biol 2020, 8, 616590.
  5. Kamola, P. J.; Kitson, J. D.; Turner, G.; Maratou, K.; Eriksson, S.; Panjwani, A.; Warnock, L. C.; Douillard Guilloux, G. A.; Moores, K.; Koppe, E. L.; Wixted, W. E.; Wilson, P. A.; Gooderham, N. J.; Gant, T. W.; Clark, K. L.; Hughes, S. A.; Edbrooke, M. R.; Parry, J. D., In silico and in vitro evaluation of exonic and intronic off-target effects form a critical element of therapeutic ASO gapmer optimization. Nucleic Acids Res 2015, 43, (18), 8638-50.
  6. <Deniz et al., 2002.pdf>.
  7. Zahradnikova, A.; Minarovic, I.; Zahradnik, I., Competitive and cooperative effects of Bay K8644 on the L-type calcium channel current inhibition by calcium channel antagonists. J Pharmacol Exp Ther 2007, 322, (2), 638-45.
  8. Cheli, V. T.; Santiago Gonzalez, D. A.; Smith, J.; Spreuer, V.; Murphy, G. G.; Paez, P. M., L-type voltage-operated calcium channels contribute to astrocyte activation In vitro. Glia 2016, 64, (8), 1396-415.
  9. Lin, J.; Taggart, M.; Borthwick, L.; Fisher, A.; Brodlie, M.; Sassano, M. F.; Tarran, R.; Gray, M. A., Acute cigarette smoke or extract exposure rapidly activates TRPA1-mediated calcium influx in primary human airway smooth muscle cells. Sci Rep 2021, 11, (1), 9643.
  10. Hashioka, S.; Klegeris, A.; McGeer, P. L., Inhibition of human astrocyte and microglia neurotoxicity by calcium channel blockers. Neuropharmacology 2012, 63, (4), 685-91.
  11. Colton, C. A.; Jia, M.; Li, M. X.; Gilbert, D. L., K+ modulation of microglial superoxide production: involvement of voltage-gated Ca2+ channels. Am J Physiol 1994, 266, (6 Pt 1), C1650-5.

Round 2

Reviewer 1 Report

The authors suitably addressed the main points raised by this reviewer. 

Reviewer 2 Report

Authors answered to all questions. Now the article is suitable for the publication